# SeafloorAI: A Large-scale Vision-Language Dataset for Seafloor Geological Survey

**Kien X. Nguyen[1], Fengchun Qiao[1], Arthur Trembanis[2], Xi Peng[1]**
[1]Deep-REAL Lab, Department of Computer and Information Sciences, University of Delaware
[2]School of Marine Science and Policy, University of Delaware
{kxnguyen,fengchun,art,xipeng}@udel.edu

## Abstract

A major obstacle to the advancements of machine learning models in marine science, particularly in sonar imagery analysis, is the scarcity of AI-ready datasets. While there have been efforts to make AI-ready sonar image dataset publicly available, they suffer from limitations in terms of environment setting and scale. To bridge this gap, we introduce `SeafloorAI`, the first extensive AI-ready datasets for seafloor mapping across 5 geological layers that is curated in collaboration with marine scientists. We further extend the dataset to `SeafloorGenAI` by incorporating the language component in order to facilitate the development of both *vision*- and *language*-capable machine learning models for sonar imagery. The dataset consists of 62 geo-distributed data surveys spanning 17,300 square kilometers, with 696K sonar images, 827K annotated segmentation masks, 696K detailed language descriptions and approximately 7M question-answer pairs. By making our data processing source code publicly available, we aim to engage the marine science community to enrich the data pool and inspire the machine learning community to develop more robust models. This collaborative approach will enhance the capabilities and applications of our datasets within both fields. Our code repository are available [1] under the CC-BY-4.0 license.

## 1 Introduction

Seafloor mapping stands at the forefront of marine science, utilizing cutting-edge technologies like multibeam echosounders and side-scan sonar to unveil the hidden complexities of the ocean floor [67, 68]. Beyond scientific research, seafloor mapping is instrumental in identifying potential resources, assessing environmental impacts, and supporting sustainable ocean management practices in the context of the blue economy [42]. However, the current analysis techniques in seafloor mapping are predominantly labor-intensive and reliant on manual interpretation by marine scientists, necessitating hundreds of hours spent meticulously examining data surveys to analyze seabed imagery [66]. This hands-on approach is not only time-consuming but also susceptible to user *subjectivity* and the limitations of individual expertise, thus introducing potential *inconsistencies* in analysis [56].

The integration of machine learning (ML) holds the promise of enhancing efficiency and reliability in seafloor mapping by automating the segmentation and classification tasks [3, 54, 38]. However, the lack of public AI-ready datasets poses a significant challenge in leveraging the full potential of AI technologies for this purpose. While there have been efforts to make AI-ready sonar image datasets publicly available, they suffer from limitations in terms of environment setting and scale. For example, the dataset in [63] was captured in a water tank, which does not accurately represent the ocean's complex conditions. Additionally, other work have only produced small-scale datasets

---

[1]https://github.com/deep-real/SeafloorAI

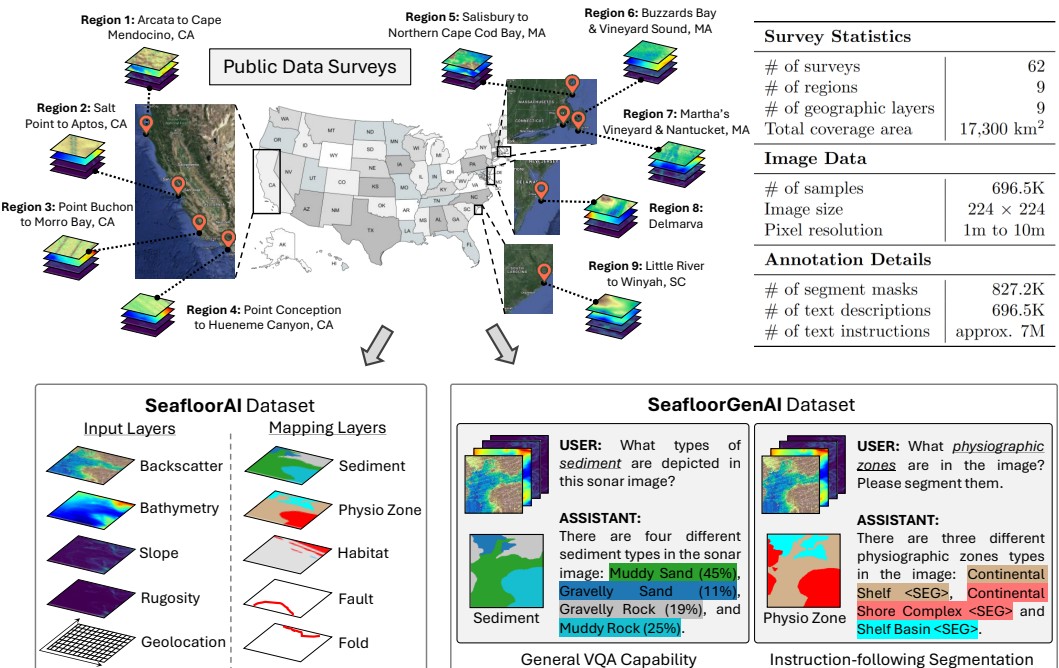

Figure 1: Overview of the spatially distributed seafloor mapping datasets. The table highlights key dataset statistics. We incorporate 62 public data surveys published by USGS and NOAA from 9 major regions to construct `SeafloorAI` and `SeafloorGenAI` datasets. Our dataset contains 9 geological layers, 4 of which are raw signals, *i.e.*, Backscatter, Bathymetry, Slope and Rugosity, and 5 annotated by human experts, *i.e.* Sediment, Physiographic Zone, Habitat, Fault and Fold. `SeafloorAI` serves as a dataset for standard computer vision tasks, *i.e.* semantic segmentation, whereas `SeafloorGenAI` constitutes a dataset for generative vision-language tasks, *i.e.*, general visual question answering and instruction-following mapping. `<SEG>` denotes the segmentation mask output by the model.

with limited area coverage [39, 61], not accounting for the generalizability of the ML models in a spatially distributed setting. On the other end, abundant public hydrographic surveys conducted by the U.S. Geological Survey (USGS) and the National Oceanographic and Atmospheric Administration (NOAA)[2] [17, 48, 49, 47, 50, 2, 5] have yet to be extensively utilized by the ML community.

To bridge this gap, we introduce `SeafloorAI`, the first extensive AI-ready sonar imagery dataset for seafloor mapping. We compiled 62 public hydrographic surveys to construct a large-scale, geo-distributed and multi-purpose dataset, with the effort to map various geological layers. Furthermore, inconsistencies in the nomenclature of geological attributes across data surveys pose a challenge on the unification and development of an extensive dataset. In collaboration with marine scientists, we have developed a framework that standardizes such nomenclature by adopting the Barnhardt classification [6] and the Coastal and Marine Ecological Classification Standard (CMECS) [1]. It guarantees uniformity throughout the dataset, enabling the evaluation of robust ML models in a spatially distributed setting. The data pool currently consists of 696K sonar images, 827K segmentation masks for 5 geological layers: Sediment, Physiographic Zone, Habitat, Fault, and Fold.

Finally, we incorporate the language component into our dataset for the development of generative vision-language models (VLMs) in marine science research. VLMs facilitate seamless interactions through textual queries and provide clear, understandable explanations throughout the analysis process [33, 34]. In addition, the ability to automate a report of the survey's findings, such as sediment composition, habitats, *etc.*, would reduce the time and effort required for manual preparation. To this end, we present a data curation pipeline that leverages both domain knowledge from marine scientists and language generation capability of `GPT-4` [46]. Specifically, we employ in-context learning [8] to generate analysis-driven question-answer pairs for each image, resulting in 7M samples and 696K language descriptions. We name the vision-language dataset `SeafloorGenAI`.

---

[2]Provides public domain data license.

| Location | Region Index | Image Resolution | Input Layers | Mapping Layers | | | | | Area (km²) |
|---|---|---|---|---|---|---|---|---|---|
| | | | | Sediment | Physio Zone | Habitat | Fault | Fold | |
| California | Region 1 | 2m/pixel | 25,817 | 25,672 | 25,823 | | | | 672 |
| | Region 2 | 2m/pixel | 123,774 | | | 123,480 | 123,774 | 123,774 | 3,148 |
| | Region 3 | 2m/pixel | 21,270 | 20,861 | 21,253 | | | | 564 |
| | Region 4 | 2m/pixel | 42,771 | | | 25,579 | 42,771 | 42,771 | 1,419 |
| Massachussetts | Region 5 | 10m/pixel | 15,827 | 4,647 | 3,387 | | | | 5,496 |
| | Region 6 | 1m/pixel | 122,441 | 122,236 | 118,175 | | | | 228 |
| | Region 7 | 1m/pixel | 1,593 | 1,507 | 1,510 | | | | 454 |
| Delmarva | Region 8 | 2m/pixel | 329,881 | | | | | | 4,525 |
| South Carolina | Region 9 | 4m/pixel | 13,141 | | | | | | 808 |
| **Total** | | | 696,515 | 174,923 | 170,148 | 149,059 | 166,545 | 166,545 | 17,314 |

Table 1: Summary of the seafloor mapping data available for each region. The input layers for sonar images include Backscatter, Bathymetry, Slope and Rugosity. Due to different mapping objectives of the original data surveys, the availability of segmentation masks is not uniform across mapping layers. Regions with unlabeled data can be utilized to pre-train the model via self-supervised learning [19].

Our contributions are summarized as follows:

1. We compile 62 public hydrographic data surveys from USGS and NOAA into a large, geo-distributed, multi-purpose and multi-modal dataset for seafloor mapping research.

2. We provide a standardization of naming convention across these surveys, under the *rigorous supervision of marine scientists*, to unify an extensive AI-ready dataset.

3. We present a data curation pipeline that produces detailed descriptions and question-answer pairs for the development of large generative vision-language models in marine science.

4. Our geo-distributed dataset contains 696K sonar images, 827K segmentation masks, 696K language descriptions and 7M question-answer pairs, covering a total area of 17,300 square kilometers.

5. We open-source our data processing code so that marine scientists could efficiently contribute their data surveys to expand the data pool.

## 2 Related Work

**Underwater Imagery Datasets.** Over the years, researchers at USGS and NOAA have carried out frequent hydrographic surveys [17, 48, 49, 47, 50, 2, 5] to collect and provide accurate and reliable information about the physical features of the water bodies and the seafloor. They are instrumental in creating accurate nautical charts to identify underwater hazards, aiding in the planning of marine infrastructure, and providing essential data for scientific research and environmental conservation. Furthermore, the data supports various economic activities, such as fishing, aquaculture, and energy production, by enabling sustainable and efficient operations.

In recent years, substantial efforts have been made to create public AI-ready underwater datasets, including forward-looking sonar (FLS), side-scan sonar (SSS), and RGB imagery. These datasets are utilized to develop machine learning models tailored for domain applications, focusing on classification or detection of geological features [3, 11, 7, 39, 54, 38] and man-made objects [77, 70, 26, 75, 32, 44, 13, 73, 72, 43, 84, 71]. Singh and Valdenegro-Toro [63] were pioneers with their FLS image dataset aimed at object detection, but their use of a controlled water tank setting may not fully reflect the complex oceanic conditions, limiting the generalizability of their results. Xie et al. [74] addressed this by extending object detection to data collected in natural water bodies, enhancing its real-world applicability. Sethuraman et al. [61] developed an SSS dataset for shipwreck detection, though its small sample size could limit model robustness. Others have also explored RGB underwater imagery for trash detection [69] and semantic segmentation [27].

Our research focuses on transforming the USGS and NOAA hydrographic surveys into a comprehensive, multi-scale, multi-purpose and multi-modal SSS imagery dataset. This initiative aims to propel advancements in both marine science and machine learning research, creating a bridge between extensive marine data resources and innovative computational techniques.

**Why side-scan sonar?** Compared to FLS and RGB imagery, SSS offers distinct advantages for underwater imagery analysis. Side-scan sonar provides a wider coverage area, and creates high-resolution images that clearly delineate the seabed texture, which is essential for geological surveys, shipwreck location, and habitat mapping. Unlike FLS, which is primarily used for obstacle avoidance, SSS offers a broad, fan-shaped beam that scans the ocean floor to either side of the towfish or autonomous underwater vehicle, capturing detailed images of the seafloor texture. Moreover, SSS is less affected by water turbidity compared to RGB cameras, which struggle with visibility in murky waters and suffer from significant color loss at depth due to light absorption. This allows SSS to produce consistent and reliable imagery under a variety of underwater conditions, where optical methods would fail. Still, SSS is only a 2D representation of the seabed. We also incorporate 3D information such as water depth to describe the underwater topography. This allows for a broad scope of underwater imagery analysis, providing robust data suitable for in-depth assessments.

**Comparison with Existing Datasets.** Our dataset is a comprehensive and expansive dataset that serves two primary purposes: (1) to act as a benchmark for various tasks and (2) to train foundation vision or vision-language models with a focus on seafloor morphodynamic analysis. In contrast to existing datasets [63, 74, 61, 69], which may specialize in single machine learning tasks or offer limited data samples, our dataset provides a diverse array of seafloor mapping tasks sourced from geographically diverse regions. Additionally, we make our data processing source code publicly available, encouraging further expansion of the dataset towards the magnitude of large-scale natural imagery datasets [62, 9, 28, 60, 59, 35].

**Datasets in other Scientific Domains.** Following the success of large foundation models in natural imagery [55, 14, 83, 82, 35, 12, 4, 79, 57, 31, 76, 81], there has been a significant push to develop expansive datasets tailored for training large foundation models for specific domain applications. In remote sensing, initiatives such as RSVQA [37], RSVQA-BEN [36], and RSGPT [23] have been developed to enhance general VQA capabilities, while MUSE [30] targets more complex reasoning tasks. Similarly, in medical imaging, datasets such as PathVQA [22], PMC-VQA [80], XrayGPT [65], LLaVA-Med [33], and OmniMedVQA [24] aim to improve the visual and textual understanding of various body parts through the analysis of MRI, X-rays, *etc.* These datasets comprise hundreds of millions of samples, posing significant acquisition challenges, particularly in marine science where data annotation is notably expensive. To address this, our initiative seeks to develop a large-scale dataset, aiming to significantly expand the resources available for marine science.

# 3 The `SeafloorAI` Dataset

## 3.1 Dataset Overview

`SeafloorAI` is a large, geo-distributed, multi-purpose dataset designed to map various geological layers of the seafloor. It is catered for training computer vision models, *i.e.* CNNs and Vision Transformers that produce semantic segmentation masks. Furthermore, it facilitates the studies of fundamental ML problems such as robust optimization [53, 51, 52, 45]. The dataset also serves as a basis for constructing the generative vision-language variant, `SeafloorGenAI`, discussed in Sec. 4.

Our dataset is compiled from 62 geological data surveys published on USGS and NOAA repositories, spanning an area of 17,300 square kilometers. This dataset features a broad geographical distribution, covering the nearshore zones of several states, including California [18], Massachusetts [49, 47, 50, 2], Delmarva [48], and South Carolina [5]. These areas are further divided into 9 regions. The data for this dataset were collected over a period spanning from 2004 to 2024, using a variety of single side-scan sonars and multibeam echosounders with different frequencies. These instruments were employed to record the texture (Backscatter) and depth (Bathymetry) of the seafloor.

The surveys have been meticulously annotated by domain experts, focusing on five key geological layers: Sediment, Physiographic Zone, Habitat, Fault, and Fold as detailed in Tab. 1. This expansive and detailed dataset provides a comprehensive view of geological and environmental features across a wide range of coastal environments. In summary, we convert the raw raster data into a large-scale machine learning-ready dataset containing 696,515 input samples, and 827,220 annotated segmentation masks across various layers.

## 3.2 Data Processing

The input layers, consisting of Backscatter and Bathymetry signals, are provided as raster data in `GeoTIFF` format. The five mapping layers serve as the ground-truth annotations, defining five tasks for the model training and evaluation. These layers come in `shapefile` format that stores the location (*i.e.*, longitude and latitude), shape (*i.e.*, polygons) and attributes of geological features (*i.e.*, sediment type). These polygons define the regions of interest on raster images, effectively delineating the boundaries of different categories that we want to segment.

Next, we present the steps for data processing at a high level, and then go further into details with each geological layer. First of all, we reproject all layers from all surveys to the WGS84 (EPSG:4326)[3] coordinate reference system. Then, we rasterize the `shapefile` to `GeoTIFF` format, effectively converting all the annotations into 2D arrays occupying the same geo-location. Finally, we use a sliding window to split the 2D raster layers into $224\times224$ patches with a step size of 56 to avoid information loss at the edges. These patches serve as the inputs and outputs for the machine learning algorithms. This process is also referred to as "patchifying".

**Input Layers: Backscatter & Bathymetry.** Backscatter in marine science refers to the amplitude of the echoes of sound waves emitted/received by a transducer that bounce off objects or the seafloor and return to the receiver. By analyzing the time it takes for the sound waves to return and their acoustic intensity, scientists and researchers can create underwater maps of the submerged terrain and identify the composition and characteristics of the seafloor, as well as the presence of underwater objects or marine life. In our dataset, we normalize the backscatter signals to the [0, 255] range, with 255 representing the nodata value. Regarding Bathymetry, we set the nodata value to be a negative number of significant magnitude, *i.e.*, -9999. Additionally, we convert Bathymetry measurements from meters to kilometers, compressing these values into a [0,1] range for normalization purposes.

We further calculate two morphologic derivatives from Bathymetry, namely Slope and Rugosity, to more comprehensively represent the topographical features of the seafloor in the input space. Slope refers to the *steepness* of the seabed, calculated as the rate of change in elevation over a given distance. It is crucial for understanding sediment transport, habitat diversity, and the stability of underwater structures. We use `GDAL` [16] implementation of the Zevenbergen & Thorne formula [78] to estimate the slope. In brevity, the formula computes the differences in elevation between a central pixel and its eight surrounding pixels for a more smoothed and stable slope estimation. Rugosity, on the other hand, measures the *roughness* or irregularity of the ocean floor. It quantifies the amount of surface area relative to a flat plane, offering vital clues about the complexity of habitats, which affects biodiversity and ecological interactions.

For each region, we resample Bathymetry, Slope and Rugosity to the Backscatter's resolution. As a result, our dataset contains a range of resolutions across regions, from 1m to 10m per pixel, enabling both coarse and fine-grained understanding of seafloor morphodynamic analysis. After patchifying the rastered map, we only keep patches where the number of nodata pixels is below 10% the number of total pixels. In the final step, we apply interpolation to fill in the missing pixels, and median filtering to reduce speckle noise. The input contains 6 channels, including these 4 layers and 2 geo-location channels (pixel-wise longitude and latitude), resulting in a dimension of $224\times224\times6$.

| | | | |
|---|---|---|---|
| **R** | Rg | Gr | **G** |
| Rs | Rm | Gs | Gm |
| Sr | Sg | Mr | Mg |
| **S** | Sm | Ms | **M** |

Figure 2: The Barnhardt classification scheme [6] is based on four end-member units: (**R**)ock, (**G**)ravel, (**S**)and, and (**M**)ud. The other twelve composite categories represent the combinations of the four units, where the dominant texture ($> 50\%$) is in upper case, and the subordinate ($< 50\%$) in lower.

**Mapping Layers: Sediment, Physiographic Zone & Habitat.** Our dataset is derived from 62 different surveys spanning both the East and West Coasts of the United States. Given the diverse origins of the data, there are inherent inconsistencies in the annotations, such as varying standards or differing vocabularies used to label the same categories. To address this, we have developed a unification process for ground-truth labels, leading to the creation of multi-class segmentation masks for Sediment, Physiographic Zone, and Habitat. This standardization process is meticulously overseen by domain experts to ensure the accuracy and quality of the annotations.

---

[3]More information at `https://docs.up42.com/data/reference/utm`.

**(1) Sediment.** Sediments on the seafloor, composed of varied particles from multiple sources, are crucial for creating habitats, indicating geological processes, and aiding in environmental and ecological research. They play a key role in resource exploration by helping to identify potential sites for natural resource extraction and in climate change studies by preserving historical climate data. Detailed seafloor mapping using sediment analysis is vital for accurate marine navigation, scientific research, and effective marine resource management. We define a unified annotation standard for the Sediment layer, following the Barnhardt classification table [6], which is a classification scheme based on four end-member units: (**R**)ock, (**G**)ravel, (**S**)and, and (**M**)ud. The other twelve composite units represent the combinations of the four units, where the dominant texture ($> 50\%$ of the area) is in upper case, and the subordinate ($< 50\%$ of the area) is in lower case, illustrated in Fig. 2. Finally, we construct semantic segmentation masks for each input patch where each pixel contains an integer value from 0 to 16, with 0 denoting the pixels without annotations.

**(2) Physiographic Zone.** By definition, a physiographic zone refers to a distinct geographical region characterized by a uniformity in topography and underlying geological structure that sets it apart from adjacent areas. These zones are typically defined based on natural landscape features, such as the configuration of the terrain, rock formations, and soil types. Classifying these zones requires the holistic understanding of multiple geological features, hence the necessity to include the bathymetric derivatives, such as Slope and Rugosity, as input. Similar to Sediment, we also define a standard for the Physiographic Zone layer. We follow the CMECS unit code for Physiographic Province which belongs in the Geoform Component [1]. There are 21 different categories for Physiographic Zone, as shown in Fig. 3.

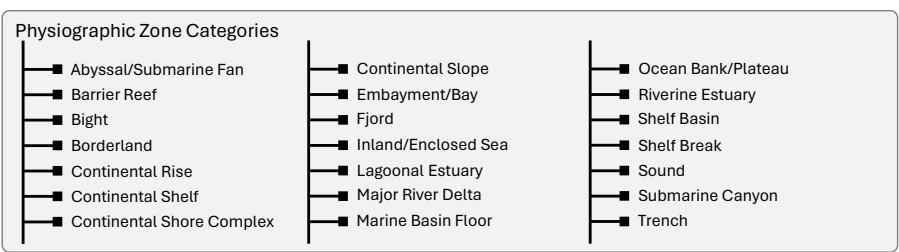

Figure 3: Twenty-one physiographic zone categories from CMECS.

**(3) Habitat.** One of the aims of seafloor mapping efforts is to delineate benthic habitats as a high-level outcome. Hall et al. [20] defined Habitat as "the resources and conditions present in an area that produce occupancy ... by a given organism." According to CMECS, a benthic habitat refers to the ecological regions at the lowest level of a body of water, including the sediment surface and sub-surface layers [1]. Benthic habitats are critical areas because they provide living space for a wide range of organisms, both flora and fauna, which are integral to the marine ecosystem. Specifically focusing on abiotic benthic habitats, these are characterized by non-living physical and chemical aspects of the environment that influence the type and abundance of organisms living there. To unify the annotations across surveys, we first gather all 144 descriptions of the polygons from the public data surveys. We then categorize these descriptions into broader groups, ultimately consolidating them into 9 distinct categories for Habitat, depicted in Fig. 4.

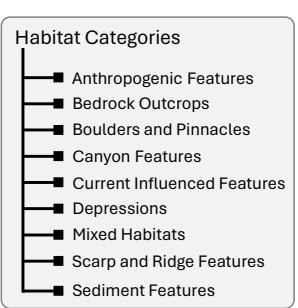

Figure 4: Nine major categories for abiotic habitat defined in `SeafloorAI`.

**Mapping Layers: Fault & Fold.** Faults and folds are significant geological features on the seafloor that are formed by tectonic movements within the Earth's crust. Faults occur when rock layers break and slide past each other due to tectonic forces, creating distinct disruptions in the seabed. Folds are bends in rock layers that occur when these layers are compressed and folded, resulting in curved or wavy stratifications. Detecting these features is crucial for understanding seismic activity and geological history of the marine environment. In our study, we formulate the binary segmentation task to identify the presence of these geological features within specific image patches, assigning the pixels containing the features a value of 1, and 0 otherwise.

# 4 The `SeafloorGenAI` Dataset

`SeafloorGenAI` incorporates vision and language understanding via visual question answering (VQA), facilitating the advancement of large vision-language models in the marine science field and the conventional studies on multi-modal learning [55, 31, 33, 41, 40]. This integration enables smooth interactions between domain experts and AI, providing clear explanations and streamlining the process of data analysis and discovery. Our dataset, consisting of 7M QA pairs and 696K language descriptions, is designed to support *general VQA capability* and *instruction-following mapping*.

**General descriptions and VQA.** Following previous work from other domains [33, 30], we utilize large language models (LLMs), specifically `GPT-4`, to generate the language descriptions and question-answer pairs for each sonar imagery sample. We employ in-context learning (ICL) [8], providing few-shot input-output pairs for the LLM. In this case, the input contains the *key analytical indicators* and the output is the description written by the marine scientists for the same image. To construct the ICL input, we, in collaboration with marine scientists, **identify** the essential information required for analysis. Subsequently, we use standard statistical and computer vision tools to **extract** three categories of information: (1) *geophysical parameters*, (2) *spatial distribution* and (3) *geological composition*. The objective is to help the model "see" the sonar image through as much detailed language descriptions as possible. For the ICL output, we ask marine scientists to manually **describe** in domain language 50 randomly selected samples from the `SeafloorAI` dataset. ICL ensures `GPT-4` can accurately mimic the domain-specific language, enhancing the quality and relevance of the generated answers. Next, we design a **prompt** to `GPT-4`, comprised of the input-output pairs and the extracted analytical indicators, to generate general descriptions and question-answer pairs for the remaining images. Finally, the domain experts carefully **evaluate** the generated language annotations to ensure quality and consistency. The last two steps form a feedback loop, creating an iterative prompt refinement process. Fig. 5 illustrates the described pipeline.

In Fig. 6, we show a sample selected from the `SeafloorGenAI` dataset. We can see that `GPT-4` is able to generate QA pairs that relate different geological layers at the same location. This helps unravel complex ecological dynamics, which is beneficial to many domain applications. We now discuss how each type of information (*i.e.* geophysical parameters, spatial distribution and geological composition) is extracted from the image.

**(1) Geophysical parameters.** These parameters are important, serving as the base for further analysis of the area. In our data processing pipeline, we employ classical analysis techniques to extract key geophysical parameters from processed data, such as water depth, mean and standard deviation of backscatter intensity, and ranges of slope, *etc.* These parameters are then systematically converted into textual format. This transformation facilitates a structured representation of complex numerical data, making it more accessible and interpretable for further analysis and reporting.

> **An example of Geophysical Parameters in the Input layers**
>
> ```
> Geolocation: (42.55°, -70.67°) to (42.53°, -70.64°)
> Depth range: -36.4 to -54.2 meters
> Backscatter mean and standard deviation: 119.7 and 72.2
> Slope range: 1.7 to 9.2 degrees
> Rugosity range: 0.01 to 0.02
> ```

**(2) Geological composition.** Understanding geological composition allows marine scientists to gain a holistic view of seafloor characteristics by examining how geological features are proportionally distributed within a specific area. Technically, this involves calculating the ratio of total pixels for each geological category relative to the overall pixels in the segmentation mask. As a result, we achieve the following:

> **An example of Geological Composition in the Sediment layer**
>
> ```
> Muddy Sand (Sm) accounts for 45% of the image.
> Muddy Rock (Rm) accounts for 25% of the image.
> Gravel Rock (Rg) accounts for 11% of the image.
> ```

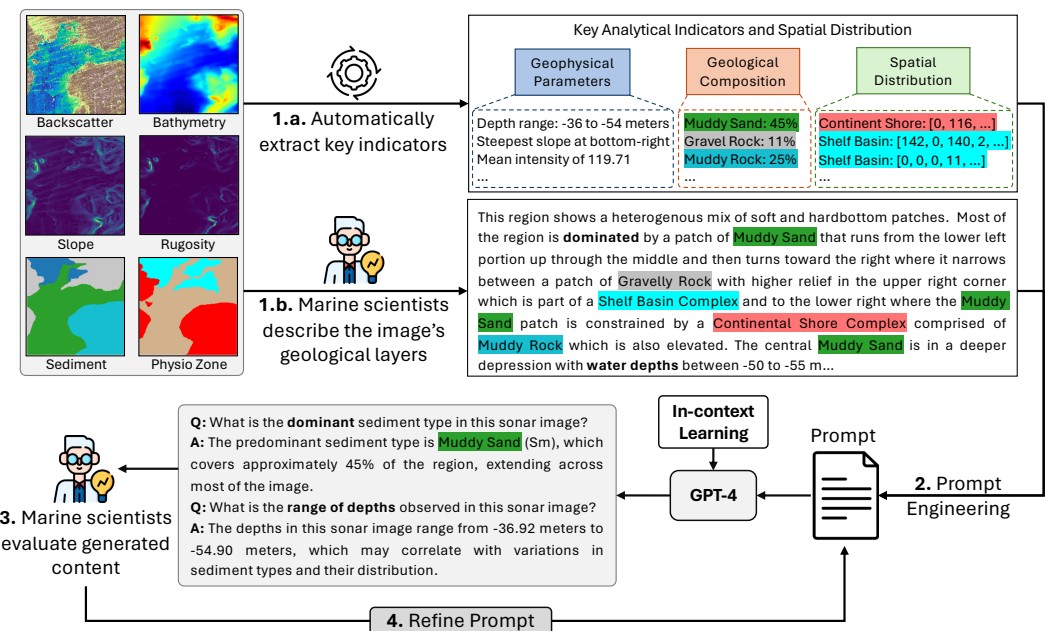

Figure 5: Pipeline for generating question-answer pairs for sonar imagery samples using GPT-4: Marine scientists first identify necessary information, followed by the extraction of *geophysical parameters*, *geological composition*, and *spatial distribution*. They then provide descriptions for a handful of samples from the SeafloorAI dataset. These description are used to design a prompt for GPT-4 to generate high-quality, domain-specific question-answer pairs, via in-context learning [8].

**(3) Spatial distribution.** Spatial distribution complements geological composition, thus giving a more comprehensive description of the image. We convert the segmentation mask of each category to polygons, which can then be fed as language into GPT-4. We first find the contours of the masks using conventional computer vision techniques, then transform them into polygon representation with the format $[x_1, y_1, ..., x_n, y_n]$, where $x_i$ and $y_i$ are the coordinates of the $i^{\text{th}}$ point in $n$ points.

---

An example of Spatial Distribution in the Physiographic Zone layer

```
Continential Shore Complex polygon at [0, 116, 0, 186, ..., 1, 117]
Shelf Basin polygon at [142, 0, 140, 2, ..., 156, 0]
```

---

**Instruction-following Mapping.** Besides VQA, we aim to equip the AI assistant with the capability to map various seafloor features across different layers in response to specific instructions. This facilitates a seamless and intuitive interaction between the AI and marine scientists, allowing for easy querying and efficient analysis. We design our dataset to be compatible with state-of-the-art VLM models, such as PixelLM [57] and LISA [31] for both single and multi-instance segmentation tasks.

---

Examples of single and multi-instance instruction-following mapping in SeafloorGenAI

```
(1) Q: Please segment [CATEGORY] in [LAYER].
    A: Sure, <SEG>.

(2) Q: What are present in the image for [LAYER]? Please segment them.
    A: [CATEGORY_1] <SEG_1>, [CATEGORY_2] <SEG_2>, ..., [CATEGORY_N] <SEG_N>.

(3) Q: Identify the areas of [CATEGORY_1] from [LAYER_1] and [CATEGORY_2]
       from [LAYER_2].
    A: Sure, [CATEGORY_1] from [LAYER_1] <SEG_1> and [CATEGORY_2]
       from [LAYER_2] <SEG_2>.
```

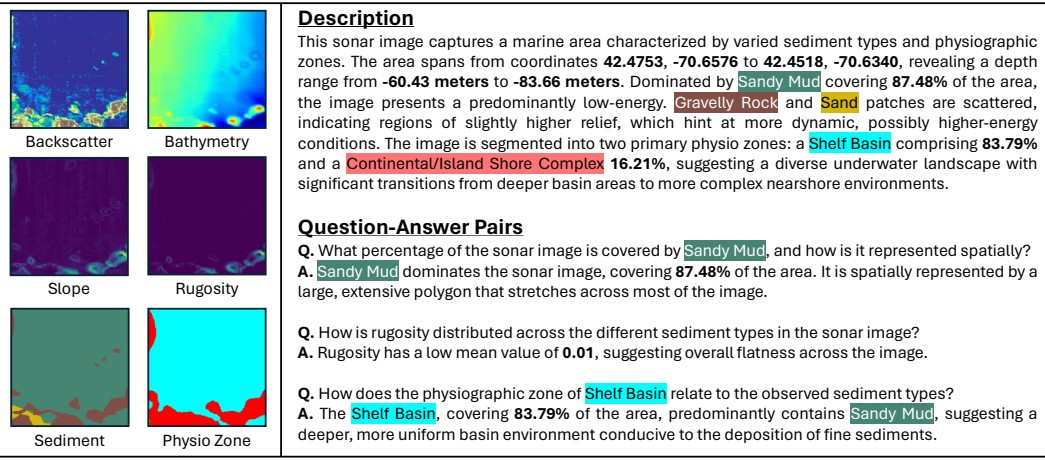

**Description**
This sonar image captures a marine area characterized by varied sediment types and physiographic zones. The area spans from coordinates **42.4753, -70.6576** to **42.4518, -70.6340**, revealing a depth range from **-60.43 meters** to **-83.66 meters**. Dominated by Sandy Mud covering **87.48%** of the area, the image presents a predominantly low-energy. Gravelly Rock and Sand patches are scattered, indicating regions of slightly higher relief, which hint at more dynamic, possibly higher-energy conditions. The image is segmented into two primary physio zones: a Shelf Basin comprising **83.79%** and a Continental/Island Shore Complex **16.21%**, suggesting a diverse underwater landscape with significant transitions from deeper basin areas to more complex nearshore environments.

**Question-Answer Pairs**
**Q.** What percentage of the sonar image is covered by Sandy Mud, and how is it represented spatially?
**A.** Sandy Mud dominates the sonar image, covering **87.48%** of the area. It is spatially represented by a large, extensive polygon that stretches across most of the image.

**Q.** How is rugosity distributed across the different sediment types in the sonar image?
**A.** Rugosity has a low mean value of **0.01**, suggesting overall flatness across the image.

**Q.** How does the physiographic zone of Shelf Basin relate to the observed sediment types?
**A.** The Shelf Basin, covering **83.79%** of the area, predominantly contains Sandy Mud, suggesting a deeper, more uniform basin environment conducive to the deposition of fine sediments.

Figure 6: An example in the `SeafloorGenAI` dataset, originated from Region 5. It features a `GPT-4` generated description and question-answer pairs designed to efficiently assist marine scientists in data analysis. The generated description covers all three key analytical indicators. Noticeably, the last QA pairs focuses on cross-layer understanding (*i.e.*, Sediment and Physiographic Zone), which is helpful for unraveling complex ecological dynamics on the seabed.

# 5 Experiments

We report some baseline experiment runs on `SeafloorAI` for multi-class segmentation. Due to space limit, we move the experiments for binary segmentation to the Supplementary Material.

**Evaluation Metrics.** We use pixel-wise accuracy (Acc), Dice coefficient (Dice) and Jaccard coefficient (mIoU) to evaluate the baseline models.

**Data Split.** We present the data splits for Sediment, Physiographic Zone and Habitat, as well as the motivation for such splits. Due to the availability of the categories in each region, we make sure that the training regions possess the set of categories that cover the testing region(s). We present our data splits for the layers in Tab. 2. For the source data, we randomly split them into 90% for training and 10% for validation. The validation set is used to select the best model for testing on the target data.

| Task | Layer | Source | Target |
|---|---|---|---|
| Multi-class Segmentation | Sediment | Region 1, Region 5, Region 6, Region 7 | Region 3 |
| | Physio Zone | Region 1, Region 3, Region 5, Region 6 | Region 7 |
| | Habitat | Region 2 | Region 4 |

Table 2: Geo-distributed data splits for the `SeafloorAI` dataset for multi-class segmentation.

**Training Details.** We employ the UNet architecture with different backbones as baselines. The UNet architecture [58] consists of a contracting path (encoder) and an expanding path (decoder), forming a U-shape. We use UNet-Base [58], UNet-ResNet18 [21] and TransUNet-ViT-B/32 [10, 15] as our baseline models for the multi-class segmentation tasks. We adopt cross-entropy as the loss function. The model was trained using the Adam optimizer [29]. The learning rate was initially set to 0.001 with a cosine annealing schedule. We use a batch size of 64 for 100 epochs, setting the patience to 5 epochs for early stopping. We perform 3 runs with different random seeds and report the model performance in Tab. 3. All runs are conducted on a single NVIDIA RTX A6000 GPU.

**Results.** Tab. 3 reports the results on the geo-distributed setting, which is similar to out-of-distribution generalization [53, 51]. We report the in-distribution (ID; on source data) and out-of-distribution (OOD; on target data) pixel-wise accuracy, Dice coefficient and Jaccard coefficient. Overall, we can see that all baseline models suffer from a signification performance degradation under distribution shift. This might be due to covariate shift (sensor types and configurations) and subpopulation shift (class imbalance). Therefore, ensuring that a model generalizes well to new, unseen distributions is a fundamental challenge. Standard training methods often assume that the training and testing data come from the same distribution, which is rarely the case in real-world applications.

| | Sediment | | | | | | | | |
|---|---|---|---|---|---|---|---|---|---|
| | Acc ID | Acc OOD | Δ Acc | Dice ID | Dice OOD | Δ Dice | mIoU ID | mIoU OOD | Δ mIoU |
| UNet-Base | 77.45 ± 0.81 | 21.49 ± 0.91 | -55.96 | 79.73 ± 0.83 | 21.59 ± 0.97 | -58.14 | 66.46 ± 1.15 | 12.29 ± 0.61 | -54.17 |
| UNet-ResNet18 | 78.45 ± 0.67 | 34.71 ± 6.79 | -43.74 | 80.78 ± 0.71 | 35.01 ± 6.86 | -45.77 | 67.90 ± 1.00 | 22.08 ± 5.73 | -45.82 |
| TransUNet | 67.90 ± 2.18 | 28.32 ± 1.04 | -39.58 | 69.94 ± 2.27 | 29.16 ± 1.05 | -40.16 | 53.98 ± 2.65 | 17.41 ± 0.71 | -36.57 |
| | Physio Zone | | | | | | | | |
| | Acc ID | Acc OOD | Δ Acc | Dice ID | Dice OOD | Δ Dice | mIoU ID | mIoU OOD | Δ mIoU |
| UNet-Base | 93.05 ± 0.16 | 56.56 ± 0.87 | -36.49 | 95.81 ± 0.18 | 57.09 ± 0.69 | -38.72 | 91.98 ± 0.32 | 43.22 ± 0.84 | -48.76 |
| UNet-ResNet18 | 92.87 ± 0.10 | 56.74 ± 2.53 | -36.13 | 95.63 ± 0.09 | 59.86 ± 2.54 | -35.77 | 91.66 ± 0.17 | 42.97 ± 3.00 | -48.69 |
| TransUNet | 90.63 ± 0.20 | 56.24 ± 1.66 | -34.39 | 93.28 ± 0.27 | 57.51 ± 1.84 | -35.77 | 87.49 ± 0.47 | 43.86 ± 2.04 | -43.63 |
| | Habitat | | | | | | | | |
| | Acc ID | Acc OOD | Δ Acc | Dice ID | Dice OOD | Δ Dice | mIoU ID | mIoU OOD | Δ mIoU |
| UNet-Base | 92.02 ± 0.18 | 70.54 ± 1.72 | -21.48 | 94.82 ± 0.20 | 71.04 ± 1.54 | -23.78 | 90.19 ± 0.37 | 56.75 ± 2.01 | -33.44 |
| UNet-ResNet18 | 92.70 ± 0.12 | 76.40 ± 1.33 | -16.30 | 95.50 ± 0.11 | 76.59 ± 1.28 | -18.91 | 91.43 ± 0.20 | 65.17 ± 1.80 | -26.26 |
| TransUNet | 88.67 ± 0.56 | 70.56 ± 0.72 | -18.11 | 91.34 ± 0.59 | 72.76 ± 0.83 | -18.58 | 84.15 ± 0.99 | 59.38 ± 1.24 | -24.77 |

Table 3: Performance of the baselines in the geo-distributed setting for multi-class segmentation.

# 6  Human Evaluation for Language Annotations

Although GPT-4 has shown strong capabilities in data annotations [64], hallucinations in LLMs are inevitable [25]. To ensure the quality of the language annotations generated by GPT-4, we describe an iterative prompt refinement process that involves human expert evaluation.

To maximize budget efficiency, we designed our procedure with several iterations of feedback and refinement. The idea is to engineer and refine our prompt to GPT-4 on a small subset of data before applying it to the whole dataset. For each iteration, (1) we annotated 1,000 random samples with GPT-4; (2) the marine scientists reviewed the quality of the generated annotations and gave feedback based on the three criteria: (i) *factual consistency* to the original annotations, (ii) *factual completeness* with respect to the analytical indicators and (iii) *coherence* to domain language; (3) we refined our prompts to GPT-4, a.k.a prompt engineering, to achieve higher quality language annotations, (4) we repeated the steps for the next iteration. Finally, when the quality is met, we will populate the entire dataset with language annotations. Due to space limitation, we include more details in the Supplementary Material.

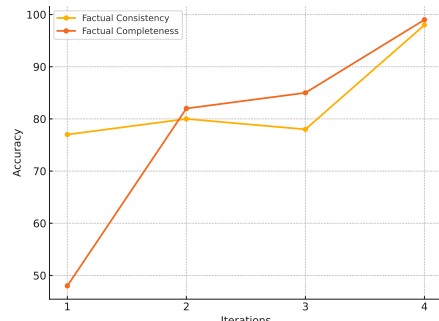

Figure 7: Accuracy for factual consistency and completeness increases over the iterations thanks to rigorous the prompt refinement procedure. GPT-4 performs worse on factual completeness potentially due to hallucinations.

# 7  Limitations and Future Work

Despite the extensiveness of our dataset, there are notable limitations to discuss. Firstly, the availability of layers is not uniform across all regions; for instance, Region 5 is missing Habitat, Fault, and Fold layers. This is due to the different mapping objectives when the data surveys were first collected. Additionally, the existing nine Habitat categories are somewhat coarse and exclude biotic classifications. We are actively collaborating with marine scientists to refine and expand the Habitat layer, making it more detailed and comprehensive. The current version of the SeafloorGenAI dataset provides annotations suitable for straightforward analytical queries and lacks the data for deeper reasoning abilities. Moving forward, we plan to enhance the dataset to support the development of reasoning-capable models similar to referring and reasoning segmentation as in [57, 31, 76], offering more profound insights into marine science questions and paving the way for data discovery. Developing this enhanced version of the dataset will require a structured and systematic approach to understanding domain-specific knowledge to accurately annotate the data. In terms of modeling, our plan for future work involves training a generative vision-language model on the SeafloorGenAI dataset, serving as a foundation ML model in marine science research.

## Acknowledgement

This work is supported by the DoD DEPSCoR Award AFOSR FA9550-23-1-0494, the NSF CAREER Award No. 2340074, the NSF SAFE Award No. 2416937, and the NSF III CORE Award No. 2412675. Any opinions, findings and conclusions or recommendations expressed in this material are those of the authors and do not reflect the views of the supporting entities.

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
