# Supplementary Material:
# SeafloorAI: A Large-scale Vision-Language Dataset for Seafloor Geological Survey

**Kien X. Nguyen[1], Fengchun Qiao[1], Arthur Trembanis[2], Xi Peng[1]**
[1]Deep-REAL Lab, Department of Computer and Information Sciences, University of Delaware
[2]School of Marine Science and Policy, University of Delaware
{kxnguyen,fengchun,art,xipeng}@udel.edu

## A   Datasheets for Datasets

### A.1   Motivation

**For what purpose was the dataset created?** The dataset was created to further advance machine learning techniques in the field of marine science.

**Who created the dataset (e.g., which team, research group) and on behalf of which entity (e.g., company, institution, organization)?** The dataset was created by the Deep-REAL and CSHEL labs at the University of Delaware. The sources of the data are from USGS and NOAA.

**Who funded the creation of the dataset?** The Department of Defense funded the project under the DEPSCoR Award.

### A.2   Composition

**What do the instances that comprise the dataset represent (e.g., documents, photos, people, countries)?** An instance is a sonar image (2D grid data), containing different geographic layers, each of which is a channel of the image.

**How many instances are there in total (of each type, if appropriate)?** Our dataset contains 696K sonar images, 827K segmentation masks, 696K general language descriptions and 7M question-answer pairs.

**Does the dataset contain all possible instances or is it a sample (not necessarily random) of instances from a larger set?** It contains all possible instances from the raw collected data.

**What data does each instance consist of?** For `SeafloorAI`, each instance, at most, consists of backscatter signal, bathymetry, slope, rugosity, geo-location, ground-truth annotations such as sediment, physiographic zone, habitat, fault and fold. For `SeafloorGenAI`, each instance also contains some question-answer pairs and an associated language description.

**Is there a label or target associated with each instance?** Not quite, there are image patches without annotations of some or all mapping layers. This is inherently due to the collected raw data.

**Is any information missing from individual instances?** No.

**Are relationships between individual instances made explicit (e.g., users' movie ratings, social network links)?** Not applicable.

**Are there recommended data splits (e.g., training, development/validation, testing)?** Yes.

**Are there any errors, sources of noise, or redundancies in the dataset?** There might be noise from the sensors when the data was collected.

38th Conference on Neural Information Processing Systems (NeurIPS 2024) Track on Datasets and Benchmarks.

**Is the dataset self-contained, or does it link to or otherwise rely on external resources (e.g., websites, tweets, other datasets)?** The raw data surveys are available on different sources. We curated and unified them into a single dataset.

**Does the dataset contain data that might be considered confidential (e.g., data that is protected by legal privilege or by doctor-patient confidentiality, data that includes the content of individuals' non-public communications)?** No.

**Does the dataset contain data that, if viewed directly, might be offensive, insulting, threatening, or might otherwise cause anxiety?** No.

**Does the dataset identify any subpopulations (e.g., by age, gender)?** No.

**Is it possible to identify individuals (i.e., one or more natural persons), either directly or indirectly (i.e., in combination with other data) from the dataset?** No.

**Does the dataset contain data that might be considered sensitive in any way (e.g., data that reveals race or ethnic origins, sexual orientations, religious beliefs, political opinions or union memberships, or locations; financial or health data; biometric or genetic data; forms of government identification, such as social security numbers; criminal history)?** No.

### A.3  Collection Process

**How was the data associated with each instance acquired? Was the data directly observable (e.g., raw text, movie ratings), reported by subjects (e.g., survey responses), or indirectly inferred/derived from other data (e.g., part-of-speech tags, model-based guesses for age or language)?** The data was observable; these were raw data collected by the USGS and NOAA, which were then processed by the authors.

**What mechanisms or procedures were used to collect the data (e.g., hardware apparatuses or sensors, manual human curation, software programs, software APIs)?** The raw data was collected by sonar sensors. We downloaded the available data from the websites through the browser.

**If the dataset is a sample from a larger set, what was the sampling strategy (e.g., deterministic, probabilistic with specific sampling probabilities)?** Not applicable.

**Who was involved in the data collection process (e.g., students, crowdworkers, contractors) and how were they compensated (e.g., how much were crowdworkers paid)?** Students were involved in the data collection process, *i.e.*, downloading data from the public data repositories.

**Over what timeframe was the data collected?** The raw data was collected from 2004 to 2024.

**Were any ethical review processes conducted (e.g., by an institutional review board)?** No.

**Did you collect the data from the individuals in question directly, or obtain it via third parties or other sources (e.g., websites)?** The data was obtained from public USGS and NOAA websites. The source of each data survey is included in our references.

**Were the individuals in question notified about the data collection?** Not applicable. The data was publicly available.

**Did the individuals in question consent to the collection and use of their data?** Not applicable. The data was publicly available.

**If consent was obtained, were the consenting individuals provided with a mechanism to revoke their consent in the future or for certain uses?** Not applicable. The data was publicly available.

**Has an analysis of the potential impact of the dataset and its use on data subjects (e.g., a data protection impact analysis) been conducted?** Not applicable.

### A.4  Preprocessing/cleaning/labeling

**Was any preprocessing/cleaning/labeling of the data done (e.g., discretization or bucketing, tokenization, part-of-speech tagging, SIFT feature extraction, removal of instances, processing of missing values)?** Reprojection, rasterization, noise removal or smoothing, patchifying, and interpolation of missing data are the major data preprocessing steps involved.

**Was the "raw" data saved in addition to the preprocessed/cleaned/labeled data (e.g., to support unanticipated future uses)?** The raw data is available online, with URLs included in the references.

**Is the software that was used to preprocess/clean/label the data available?** Yes, we also make our processing code available so that people could contribute to the data pool.

### A.5 Uses

**Has the dataset been used for any tasks already?** No.

**Is there a repository that links to any or all papers or systems that use the dataset?** No.

**What (other) tasks could the dataset be used for?** Currently, the data can only be used for semantic segmentation, but the unlabeled data can be used to pre-trained a model via self-supervised learning.

**Is there anything about the composition of the dataset or the way it was collected and preprocessed/cleaned/labeled that might impact future uses?** No.

**Are there tasks for which the dataset should not be used?** No.

**Any other comments?** No.

### A.6 Distribution

**Will the dataset be distributed to third parties outside of the entity (e.g., company, institution, organization) on behalf of which the dataset was created?** The dataset is going to be publicly available to all.

**How will the dataset will be distributed (e.g., tarball on website, API, GitHub)?** GitHub is going to be the main directory for the dataset, containing documentation, code, and URLs to where the data is stored.

**When will the dataset be distributed?** Before Dec 9th, the date of the conference.

**Will the dataset be distributed under a copyright or other intellectual property (IP) license, and/or under applicable terms of use (ToU)?** It will be distributed under the CC-BY 4.0 license.

**Have any third parties imposed IP-based or other restrictions on the data associated with the instances?** No.

**Do any export controls or other regulatory restrictions apply to the dataset or to individual instances?** No.

### A.7 Maintenance

**Who will be supporting/hosting/maintaining the dataset?** The dataset will be hosted on Zenodo. The first author will be maintaining and updating the dataset.

**How can the owner/curator/manager of the dataset be contacted (e.g., email address)?** The contact information is included in the author list.

**Is there an erratum?** No.

**Will the dataset be updated (e.g., to correct labeling errors, add new instances, delete instances)?** The dataset will continue to be expanded in the future.

**If the dataset relates to people, are there applicable limits on the retention of the data associated with the instances (e.g., were the individuals in question told that their data would be retained for a fixed period of time and then deleted)?** Not applicable.

**Will older versions of the dataset continue to be supported/hosted/maintained?** Yes.

**If others want to extend/augment/build on/contribute to the dataset, is there a mechanism for them to do so?** Yes, we will make our code publicly available so other people can contribute.

| Geological Feature | Definition |
|---|---|
| Backscatter | The measure of the intensity of sound waves or radar signals reflected back from the seabed or other underwater objects. Used to infer bottom type and other properties. |
| Bathymetry | The measurement of the depth of water bodies; essentially the underwater equivalent of topography, mapping the shape and depth of the ocean floor. |
| Slope | The steepness or incline of the seabed, calculated as the change in elevation over a specific distance. Used in predicting sediment transport and erosion. |
| Rugosity | The measure of the roughness or irregularity of the seafloor's surface texture, often used in ecological studies to assess habitat complexity. |
| Sediment | Particulate organic and inorganic material that settles at the bottom of a water body. Its composition can vary widely from clay to gravel. |
| Physiographic Zones | Broad geographic regions characterized by a distinctive landscape and geological features, which are often similar in terms of their formation and development. |
| Habitat (Abiotic) | The non-living environmental factors that influence ecosystem structures, such as water depth, temperature, and salinity. |
| Faults | A fracture or zone of fractures between two blocks of rock, which can lead to relative movement and displacement that is often measurable. |
| Folds | A bend or series of bends in rock strata or other planar surfaces, resulting from compressive forces that cause the material to buckle and deform permanently. |

Table 1: Definition of geological features that involve in our seafloor mapping dataset.

| Region Index | Geological Data Surveys |
|---|---|
| Region 1 | Offshore of Arcata, CA [6], Eureka, CA [7], Eel River, CA [5], and Cape Mendocino, CA [4] |
| Region 2 | Offshore of Salt Point, CA [34], Fort Ross, CA [33], Bodega Head, CA [32], Tomales Point, CA [35], Point Reyes, CA [68], Drakes Bay, CA [69], Bolinas, CA [15], San Francisco, CA [17], Pacifica, CA [20], Half Moon Bay, CA [11], San Gregorio, CA [12], Pigeon Point, CA [18], Scott Creek, CA [14], Santa Cruz, CA [13], and Aptos, CA [16] |
| Region 3 | Offshore of Point Buchon, CA [9], Point Estero, CA [10], and Morro Bay, CA [8] |
| Region 4 | Offshore of Point Conception, CA [31], Gaviota, CA [30], Refugio Beach, CA [25], Coal Oil Point, CA [26], Santa Barbara, CA [24], Carpinteria, CA [27], Ventura, CA [29], and Hueneme Canyon, CA [28] |
| Region 5 | Inner Continental Shelf from Nahant to Northern Cape Cod Bay, MA [65], and from Salisbury to Nahant, MA [66] |
| Region 6 | Buzzards Bay and Vineyard Sound, MA [1] |
| Region 7 | Inner Continental Shelf of Martha's Vineyard from Aquinnah to Wasque Point, and Nantucket from Eel Point to Great Point, MA [67] |
| Region 8 | NOAA Survey H11554 [38], H11555 [39], H11647 [40], H11648 [41], H11649 [42], H11650 [43], H11872 [44], H11873 [45], H11874 [46], H11992 [47], H12001 [48], H12002 [49], H12003 [50], H12091 [51], H12092 [52], H12093 [53], H12094 [54], H12160 [55], H12161 [56], H12336 [57], H12337 [58], H12338 [59], H12339 [60], H12394 [61], H12395 [62], H12396 [63], and H12397 [64] |
| Region 9 | Long Bay from Little River to Winyah Bay, SC [2] |

Table 2: Region Index and its corresponding hydrographic surveys in the dataset. Our dataset includes 62 surveys across 9 regions on both the U.S coasts.

## B   Appendix

### B.1   Glossary for Geological Features

We provide the definition of the geological features covered in the paper in Tab. 1.

### B.2   Details on public data surveys

Table 2 includes the name of the original 62 USGS/NOAA data surveys for each region. Our data pool is going to be expanded further as more data sources are available for processing.

**Discussion on geographical diversity.** Despite consisting only 9 regions on the U.S coasts, `SeafloorAI`, which we see as a starting point towards a goal of more extensive global settings, is still very representative of geologically diverse environments. We have data gathered from both an active (West Coast) and passive (East Coast) plate boundary with varied tectonic history and lithologic composition. Furthermore, within the east coast dataset we have a range spanning from the Middle Atlantic bight with a wide continental shelf all the way up into New England which includes direct glacial impacts on the coastline and seabed. The areas represent similar settings around the world [23, 19]. For instance the East Coast of the US as an Amero-trailing edge tectonic setting is very similar to that found off the east coast of South America. The West Coast of the US, which is an active plate boundary with a narrow shelf, is also similar to those found along the west coast of South American and other similar plate boundary settings. There are admittedly more geological settings that we have hopes of gathering but we must work first with what is available and in this paper we seek to develop and demonstrate the effectiveness of the framework that could be acquired readily at this time. Our goal is to develop and test this framework and by illustrating the effectiveness and benefits in using this framework to encourage further adoption to expand the representative datasets.

## B.3 Formulas for Bathymetric Derivatives

**Zvenbergen & Throne Slope Formula.** Slope, or *steepness*, of the terrain, can be computed as the gradients of the elevation. For a pixel at $(i, j)$ in a digital elevation model (DEM) or the bathymetry raster file in our case, $Z_{i,j}$ denotes the elevation at that pixel. The slope $\theta$ is calculated using the following steps:

1. Compute the finite differences:

The elevation differences along the $x-$direction $(\partial Z/\partial X)$ and $y-$direction $(\partial Z/\partial Y)$ are computed by averaging the differences between the central pixel and its eight immediate neighbors, equivalent to the locality of a $3 \times 3$ sub-matrix:

$$\frac{\partial Z}{\partial X} = \frac{Z_{i-1,j-1} + 2Z_{i-1,j} + Z_{i-1,j+1} - Z_{i+1,j-1} - 2Z_{i+1,j} - Z_{i+1,j+1}}{8 \times \text{pixel resolution}} \quad (1)$$

$$\frac{\partial Z}{\partial Y} = \frac{Z_{i-1,j-1} + 2Z_{i,j-1} + Z_{i+1,j-1} - Z_{i-1,j+1} - 2Z_{i,j+1} - Z_{i+1,j+1}}{8 \times \text{pixel resolution}} \quad (2)$$

2. Compute slope:

The slope $\theta$ is then calculated using the formula:

$$\theta = \arctan\left(\sqrt{\left(\frac{\partial Z}{\partial X}\right)^2 + \left(\frac{\partial Z}{\partial Y}\right)^2}\right) \quad (3)$$

Finally, we convert the result to degrees: $\theta_{\text{deg}} = \theta \times \frac{180}{\pi}$

**Rugosity Formula.** Rugosity, or *roughness*, of an area can be calculated by taking the ratio of the surface area over the planar area. Given a DEM, the terrain is represented by a grid of elevation points, where each point is denoted by $Z$. The slope at each point affects how much larger the actual surface area of that point is compared to its projection on a horizontal plane.

For a small surface element with sides $dX$ and $dY$ in the horizontal plane, the actual surface area $dA$ of this element on the terrain, which is not flat due to the slope, can be derived from the gradient of the elevation. Here are the steps to calculate the actual surface area:

1. Calculate the gradients:

Similar to the slope formula above, the gradient of the elevation $Z$ at any point is given by the vector $\nabla Z = \left(\frac{\partial Z}{\partial X}, \frac{\partial Z}{\partial Y}\right)$.

2. Compute the differential surface element:

The differential surface element $dA$ on a sloped surface can be calculated from the magnitudes of the partial derivatives:

$$dA = \sqrt{1 + \left(\frac{\partial Z}{\partial X}\right)^2 + \left(\frac{\partial Z}{\partial Y}\right)^2} \, dXdY \quad (4)$$

3. Integrate over the entire area:

We integrate $dA$ over the area of interest to find the total surface area $A$ of the terrain:

$$A = \iint \sqrt{1 + \left(\frac{\partial Z}{\partial X}\right)^2 + \left(\frac{\partial Z}{\partial Y}\right)^2} \, dX \, dY \tag{5}$$

4. Derive rugosity:

Finally, we take the ratio of the surface area $A$ over the planar area to derive Rugosity:

$$R = \frac{A}{\text{pixel resolution}^2} \tag{6}$$

### B.4 More on Human Evaluation for Language Annotations

In this section, we provide more details on the iterative human expert evaluation and prompt refinement procedure for language annotations. We define the three criteria for evaluation as follows:

1. **Factual consistency**: the information of geological features contained in the descriptions is consistent with the provided analytical indicators.
2. **Factual completeness:** the coverage of geological features in the descriptions with respect to the provided analytical indicators.
3. **Coherence:** the logical and fluid integration of information in the text, allowing it to be easily understood by the marine scientist. This metric is qualitative.

We present our results on *factual consistency* and *factual completeness* across four iterations, and explain how feedback from marine scientists on *coherence* informed our prompt refinements for GPT-4. Overall, GPT-4 demonstrated high and stable performance in factual consistency, indicating its ability to accurately reproduce the facts provided in the prompt.

However, the model initially struggled with factual completeness due to its tendency to hallucinate and deviate from the main focus of the task instructions, necessitating iterative prompt refinements. In the first iteration, we employed a simple prompt that included only the key analytical indicators and task instructions. The outputs generated by GPT-4 lacked coherence and omitted many details from the indicators, resulting in an average factual completeness score of 48%.

To address this issue, we explicitly added a task rule to our prompt, instructing GPT-4 to include all the analytical indicators in the generated descriptions. This modification significantly enhanced performance, with factual completeness scores increasing to 82%, 85%, and 96% in the subsequent iterations.

Regarding coherence, we updated the prompt by leveraging qualitative feedback from the marine scientists, who noted issues such as "the output is not eloquent" and "the output reads more like a list." To address these concerns in subsequent iterations, we incorporated in-context learning (ICL) [3], which provides the LLM with example inputs and outputs for reference.

To construct these examples, we randomly selected 50 samples from the SeafloorAI dataset encompassing various categories. We then asked the scientists to write detailed descriptions based on the input layers (e.g., backscatter, bathymetry) and the annotated segmentation masks of the mapping layers. When generating language annotations for new samples, we included the analytical indicators as the ICL input and the manually written descriptions as the ICL output, as illustrated in Section B.5. This approach helps the LLM mimic domain-specific language, resulting in more coherent and domain-aligned descriptions.

### B.5 Sample Prompt for GPT-4

In this section, we present a sample prompt used with GPT-4. Briefly, the prompt's structure includes several in-context learning examples, detailed information about the analytical indicators of the data sample we are annotating, clear task instructions, and specific rules designed to guide the LLM's output according to our desired objectives.

The template of a prompt input to GPT-4 to generate QA pairs

Here are some pairs of example containing the key analytical indicators of a
sonar image patch and its corresponding domain expert's description:

1. <Input: Analytical Indicators> <Output: Expert's Description>
2. <Input: Analytical Indicators> <Output: Expert's Description>
3. ...

Here are some key analytical indicators extracted from a 224 x 224 sonar
image. This patch contains 4 input layers (backscatter, bathymetry, slope and
rugosity) and <# of mapping layers> mapping layers (sediment/physio zone/
habitat/fault/fold):

<Input: Analytical Indicators for new sample>

There are two tasks for you:
(1) Generate a short description describing the geophysical parameters,
geological composition and spatial distribution of attributes in the image.
(2) Generate 10 question-answer pairs focusing on (sediment/physio zone/
habitat) composition, spatial distribution (based on the provided polygons),
and general analytical indicators, based on all the information provided.

The generated description and question-answer pairs MUST follow the
requirements:
- The description MUST refer to what is available in the provided information
and MUST NOT refer to related concepts outside of the scope.
- The description MUST contain ALL information of the provided geological
attributes.
- The description MUST NOT be too broad and must be meaningful.
- The answers MUST be concise.
- The output MUST follow the wording style of the samples from the domain
experts provided at the beginning.

---

A sample of in-context learning input-output pair for prompting GPT-4

Input:
Depth range: -36.92 meters to -55.00 meters
Backscatter statistics: mean of 119.71, standard deviation of 72.02
Sediment composition:
- Muddy sand (Sm): 45%
- Gravelly sand (Sg): 9%
- Gravelly rock (Rg): 20%
- Muddy rock (Rm): 26%
Sediment polygons: the polygon format is [x1, y1, x2, y2, ..., xn, yn] for n
points. The following is the name of the sediment type followed by the
polygons of each connected component of that sediment in the image.
- Muddy Sand: [223, 50, 79, 74, ..., 223, 50]
- Gravelly Sand: [91, 48, 100, 53, ..., 91, 48], [0, 0, 37, 63, ..., 0, 0]
- Gravelly Rock: [3, 114, 0, 179, ..., 3, 114], [0, 2, 6, 87, ..., 0, 2], ...
- Muddy Rock: [222, 100, 96, 136, ..., 222, 100]
Physiographic Zone polygons:
- Continental/Island Shelf: [223, 0, 123, 43, ..., 223, 0]
- Continental/Island Shore Complex: [0, 116, 1, 186, ..., 0, 116], ...
- Shelf Basin: [189, 7, 110, 18, ..., 189, 7], [0, 11, 51, 68, ..., 0, 11]
------------------------------------------------------------------------
Output:
This region shows a heterogenous mix of soft and hardbottom patches. Most of
the region is dominated by a patch of muddy sand that runs from the lower
left portion up through the middle and then turns toward the right where it

narrows between a patch of gravelly rock with higher relief in the upper
right corner which is part of a shelf basin complex and to the lower right
where the muddy sand patch is constrained by a continental shore complex
comprised of muddy rock which is also elevated. The central muddy sand is
in a deeper depression with water depths ranging between 50-55 m whereas the
adjacent gravelly rock and muddy rock areas have a shallower depth range of
between 36-40 m depth. Backscatter within the muddy zones is much lower with
ranges of around 25-120 compared to the areas with gravelly rock and muddy
rock which possess much higher backscatter in the range above 150. The slope
and rugosity is very low in the muddy zones illustrating that these deeper
areas with fine muddy sediment are flat and seemingly featureless. Whereas,
the edges of the gravelly rock and muddy rock zones are well delineated by
areas of high slope and rugosity which represent the shift from a deeper
flatter zone to the shallower and more rugged terrain in these zones.

## B.6 Additional Examples of the `SeafloorGenAI` Dataset

Please refer to Fig. 1 for additional examples in the `SeafloorGenAI` dataset. We include the general
descriptions along with some analysis-driven question-answer pairs.

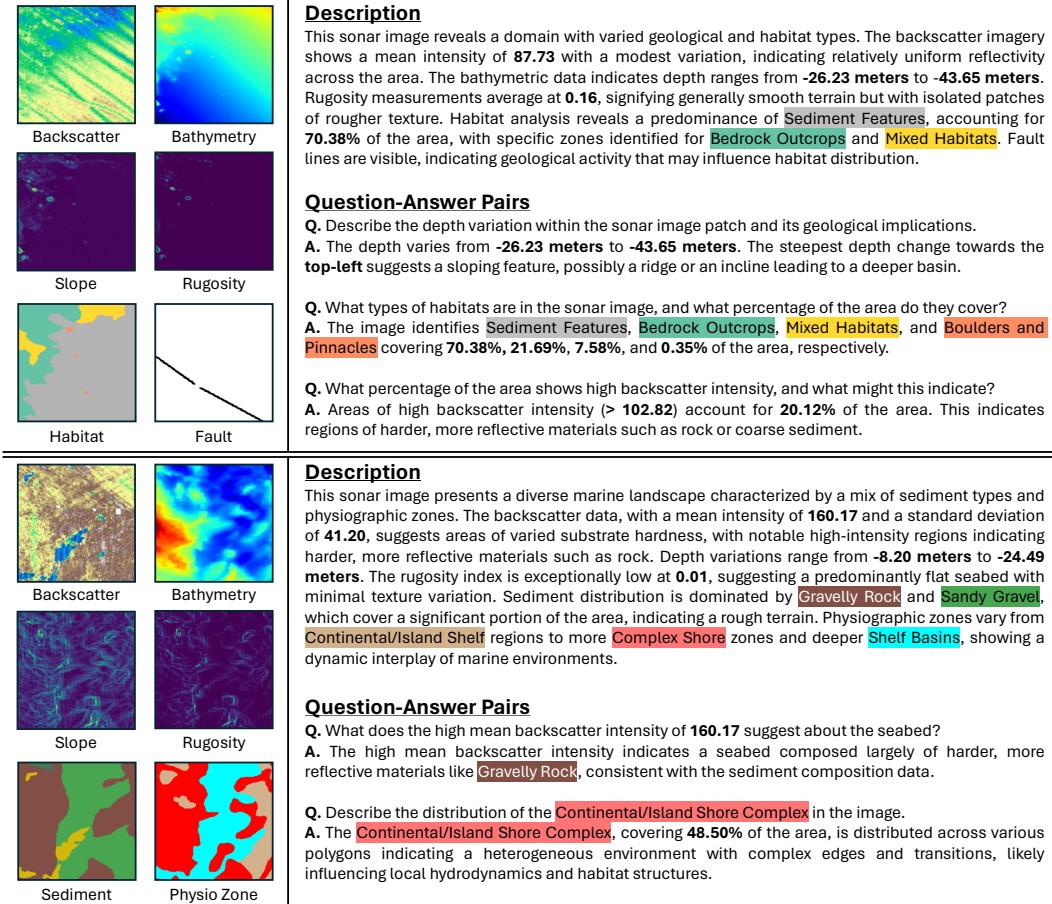

Figure 1: Two additional examples in the `SeafloorGenAI` dataset, originated from Region 2 (**top**)
and 5 (**bottom**). It features a `GPT-4` generated description and question-answer pairs designed to
efficiently assist marine scientists in data analysis. The generated description covers all three key
analytical indicators.

## B.7 More Experiments

We report some more baseline experiment runs on the binary segmentation tasks on the Fault and Fold layers in this section.

**Evaluation Metrics.** We use Dice coefficient and mean IoU for binary segmentation.

**Data Split.** Binary segmentation often comes with the problem of class imbalance where the number of background pixels overwhelms that of the object of interest. We choose Region 2 to be the source and Region 4 as target data because Region 2 has more data (Tab. 3). We also employ sample weighting during the training process to alleviate the problem.

| Task | Layer | Source | Target |
|------|-------|--------|--------|
| Binary Segmentation | Fault | Region 2 | Region 4 |
|  | Fold | Region 2 | Region 4 |

Table 3: Geo-distributed data splits for the binary segmentation tasks on the `SeafloorAI` dataset.

**Training Details.** We employ the UNet architecture with different backbones as baselines. For binary segmentation, we use more complex backbones, i.e. UNet-ResNet50 [21] and UNet-DenseNet121 [22], due to the problem of severe class imbalance.

We adopt Focal loss [37] combined with Dice loss for binary segmentation; we set the Focal loss's $\gamma = 5.0$. For binary segmentation, we also employ sample weights sampling to alleviate class imbalance. The weight of each sample with the label of interest (labeled 1) is inversely proportional to the ratio of the number of the foreground pixels to the total number of pixels. The samples with all background pixels are assigned very low weights. The model was trained using the Adam optimizer [36]. The learning rate was initially set to 0.001 with a cosine annealing schedule. We use a batch size of 32 in binary segmentation tasks for 100 epochs, setting the patience to 5 epochs for early stopping. We perform 3 runs with different random seeds and report the model performance in Tab. 4. All runs are conducted on a single NVIDIA RTX A6000 GPU.

**Results.** Tab. 4 reports the results on the geo-distributed setting. Overall, we can see that all baseline models suffer from a signification performance degradation under distribution shift, similar to the multi-class segmentation setting.

|  | **Fault** | | | | | |
|---|---|---|---|---|---|---|
|  | Dice ID | Dice OOD | $\Delta$ Dice | mIoU ID | mIoU OOD | $\Delta$ mIoU |
| UNet-DenseNet121 | $90.29 \pm 0.10$ | $57.42 \pm 0.92$ | -32.87 | $82.32 \pm 0.47$ | $52.69 \pm 0.44$ | -29.63 |
| UNet-ResNet50 | $92.64 \pm 0.54$ | $57.18 \pm 0.63$ | -35.46 | $86.24 \pm 0.72$ | $53.10 \pm 0.23$ | -33.14 |
|  | **Fold** | | | | | |
|  | Dice ID | Dice OOD | $\Delta$ Dice | mIoU ID | mIoU OOD | $\Delta$ mIoU |
| UNet-DenseNet121 | $88.84 \pm 0.32$ | $54.23 \pm 0.54$ | -34.61 | $79.94 \pm 0.56$ | $45.89 \pm 0.54$ | -34.05 |
| UNet-ResNet50 | $92.43 \pm 0.33$ | $55.77 \pm 0.67$ | -36.66 | $85.93 \pm 0.64$ | $47.02 \pm 0.22$ | -38.91 |

Table 4: Performance of the baselines in the geo-distributed setting for binary segmentation.