# OpenReview forum: "SeafloorAI: A Large-scale Vision-Language Dataset for Seafloor Geological Survey"
_NeurIPS.cc/2024/Datasets_and_Benchmarks_Track — NeurIPS 2024 Track Datasets and Benchmarks Poster_

### Official Review · Reviewer_AnUV · 2024-07-19
**Good paper about a new large scale dataset for seafloor marine science**

**Rating:** 7
**Confidence:** 5

**Review:**

This paper is high quality, high clarity, and highly original. It is significant to the marine science and intersection with machine learning applications for sonar imaging, as it is currently the largest sonar image dataset available with useful labels.

The pros are the large dataset size, the dataset collection is correct (using publicly available data from multiple sources), and the tasks, while simple, do make sense for the scale of the task (oceanic level), and finally, there are multiple modalities which could enable multimodal models in oceanographic data.

The cons are the limited selection of modalities (not really possible to change at this scale), and the generation of question-answer pairs using GPT4, which might not have been human validated, and no dataset samples were available during the review process (only the very limited selection present in the paper and supplementary). The Zenodo link did not work for me due to permission issues.

**Strengths:**

- The proposed dataset SeafloorAI contains data from multiple geographical regions, covering a large seafloor area, ensuring diversity.
- This is by far the largest dataset of sonar images that is made publicly and for machine learning use.
- The dataset contains multiple modalities, including sidescan sonar, bathymetry, geolocalization, and some derived features like slope and rugosity.
- The dataset covers many possible use cases, the paper only presents segmentation of several seafloor features (sediment, habitat, faults, fold, and physiozones) and visual question answering, and other tasks like self-supervised learning are possible at this scale.
- The SeafloorGenAI dataset contains 7M question-answer pairs, that were automatically generated using GPT4, the scale is massive, and could be useful to build chatbots for marine scientists.
- The supplementary contains preliminary/first results for segmentation tasks, which add a layer of validation that the SeafloorAI dataset is useful, even containing significant distribution shift that could be used for future development and evaluation of machine learning models.

**Additional Feedback:**

Just one question, how were the labels for the SeafloorGenAI dataset validated? Did humans look at each of the 7 million QA pairs?

**Clarity:**

Yes, the paper is very well written, figures are clear, I have no feedback on writing.

**Correctness:**

Yes, the paper is correct, and the dataset is built in a sound way. There are not many alternatives for building sonar image dataset for marine and underwater applications.

**Documentation:**

The supplementary material contains most of the required information, but the URL for reviewer access did not work for me due to Zenodo permission issues, this might have been an oversight by the authors, so this reviewer could not examine the dataset in detail. Other components of documentation seem to be described well.

**Ethics:**

No ethics issues

**Limitations:**

Yes, the paper contains an excellent limitations section, covering limitations on geographical coverage of the dataset, distribution shifts, and not having trained a model on the SeafloorGenAI dataset due to size.

**Opportunities For Improvement:**

- The labels for the VLM dataset were generated using GPT4, while this is not a problem by itself, it is not clear form the paper how the question-answer pairs and prompts were validated. There are 7 million QA pairs, how were they validated? The paper claims human validation but does not confirm that all 7 million QA pairs were validated, and they were generated from 50 in-context learning human made samples.
- The dataset or a sample of it, was not made available during the review process, I recognize there is an anonymous link to code, but the Zenodo link to the dataset did not work for me, due to permission issues. It would have been nice to examine some dataset samples during review.

**Relation To Prior Work:**

Yes, the paper discusses previous ML-focused datasets for sonar images and marine applications, and datasets of similar scale in other scientific domains. The paper is very well connected to the literature.

**Summary And Contributions:**

This paper describes a new dataset of sidescan sonar images for seafloor mapping, consisting of 700K images, for different vision and vision-language tasks. The dataset is divided into a vision dataset (SeafloorAI) and a vision-language dataset (SeafloorGenAI) with question-answer pairs. The data is aggregated from public datasets from USGS and NOAA, covering multiple coastal regions in the US. The data consists of patches from sidescan sonar images, bathymetry, and some 3D features of the seafloor.

Contributions are:
- The datasets SeafloorAI and SeafloorGenAI, being the largest public dataset of sonar images in underwater scenes (seafloor) to date, with around 700K image patches.
- Data sources were merged by harmonizing labels and conventions, to make a single unified dataset.
- Question-answer pairs and prompts generated using GPT4 to build a chatbot usable for marine science purposes.

---

> ### Author Rebuttal · Authors · 2024-08-16
>
> We thank the reviewer for the valuable comments. Below are our responses to the points raised by the reviewer. We hope these will help improve the clarity of the paper.
>
> **1. How are the language annotations validated?**
> In the language annotation pipeline, we asked the marine scientists to validate a thousand samples generated by GPT-4, especially the generated language descriptions. The experts are tasked to validate the descriptions based on two criteria: (1) consistency to the original annotations and (2) coherency to the domain language. Currently, we are on the last phases of quality assessment and will include the details on the evaluation pipeline as a separate section in the Supplementary Materials. We further describe our language annotation procedure as follows:
>
> Due to budget constraints on the usage of OpenAI API, we meticulously designed our procedure with multiple iterations. The idea is to engineer our prompt to GPT-4 on a small subset of data before applying it to the whole dataset. For each iteration, (1) we annotated a thousand random samples with GPT-4, (2) the marine scientists reviewed the quality of the generated annotations and gave feedback based on the two mentioned criteria, (3) we refined our prompts to GPT-4, a.k.a prompt engineering, to achieve higher quality language annotations, (4) we repeated the steps for the next iteration. Finally, when the quality is met, we will populate the entire dataset with language annotations.
>
> To prevent hallucination in large language models, we made sure to provide GPT-4 as much textual information extracted from the annotations as possible. An example can be found here: https://anonymous.4open.science/r/SeafloorFM/sample_prompt.txt
>
> **2. Working Zenodo link.** We apologize for the oversight on the Zenodo link. We have fixed the issue and provide the working link here: https://zenodo.org/records/11630750. These are the samples from Region 5 in the paper. In the anonymous git repository (https://anonymous.4open.science/r/SeafloorFM), we also added a script to visualize the data (``visualize_vision_data.py`` and ``visualize_vlm_data.py``) and some visualization of the data (under  ``vis-vision/`` and ``vis-vlm/`` directories) for the reviewers’ convenience (as the data for this region takes up 40GB of storage). The model training code is also included.

---

> > ### Author Response · Authors · 2024-08-23
> > **We would like to hear back from Reviewer AnUV**
> >
> > Dear Reviewer AnUV,
> >
> > We kindly request your feedback on our rebuttal. We've addressed key points including: our language annotation validation process, providing a working dataset access link, and our prompt engineering approach to prevent hallucinations. We appreciate your time and look forward to your thoughts on these clarifications.
> >
> > Best,\
> > Authors

---

> > > ### Comment · Reviewer_AnUV · 2024-08-23
> > > **Thanks for rebuttal**
> > >
> > > Dear Authors, thanks for the detailed rebuttal. I agree that the Zenodo link now works and I can see samples of the dataset.
> > >
> > > I can increase my score slightly, but still I believe the annotation validation process could lead to problems, looking at 1K annotations is not enough when millions of samples have been generated. Unfortunately I do not have a easy solution, the authors should consider if in the future some labels are found to be incorrect, then this would greatly decrease the dataset value.

---

> > > > ### Author Response · Authors · 2024-08-26
> > > >
> > > > Dear Reviewer AnUV,
> > > >
> > > > We acknowledge that one thousand samples used for human evaluation might not be sufficient. Therefore, we are currently conducting the evaluation procedure via multiple iterations to ensure the highest quality for the generated data, as mentioned in our rebuttal, with the plan to increase our manpower in the future.
> > > >
> > > > We are currently looking to increase our team by involving more additional marine scientists, which will enable us to cover a more substantial portion of the dataset with a higher degree of scrutiny. This expansion is not just about increasing numbers; each scientist brings unique expertise that is crucial for identifying nuanced inaccuracies that might be missed in a less diverse review panel.

---

### Official Review · Reviewer_X3ie · 2024-07-23
**SeafloorGenAI Review**

**Rating:** 8
**Confidence:** 4

**Review:**

Overall, this work is of extremely high quality and originality. There are a small number of issues with clarity, but these can be easily resolved during the review process with only minor changes to the paper. While this paper may not be significant to the broader machine learning community, it has the potential to have a large impact on a number of sciences and industries, including United Nations (UN) Sustainable Development Goals (SDG) 6 (clean water), 13 (climate action), and 14 (life below water). Listed below are particular pros and cons of this paper:

Pros:

* Dataset is large, multi-modal, multi-purpose, and constructed with care
* Dataset is rather unique, making it more impactful and less likely to be superseded by another dataset in 6 months
* Authors directly collaborate with marine scientists to ensure the usefulness of the dataset
* Paper is extremely well written and flows well

Cons:

* Dataset is not particularly useful to the broader machine learning community
* Geographic diversity of the dataset is limited to nearshore regions and to the US

**Strengths:**

This paper is extremely well written, with almost all of my questions being answered by the end of the paper. The paper flows well, and only introduces detailed concepts once they need to be known. The paper is virtually free of typos and grammar mistakes, showing how much work the authors put into polishing the paper for this submission.

The authors do a good job of introducing why such a dataset is important, demonstrating how it can be used for benchmarking of existing models, pre-training of new foundation models, and interactive labeling and question answering by domain experts. They also explain why existing data surveys are often insufficient, highlighting the subjectivity of human labeling and differences in labeling standards. Furthermore, they stress several times throughout the paper that this work was done under the supervision and guidance of marine science experts, something this reviewer considers absolutely necessary for interdisciplinary research.

While this dataset is somewhat limited to a "niche" application area (marine science) and not useful to the broader machine learning community, the potential impact of ocean stewardship far outweighs other "yet another benchmark"-style submissions so prevalent at NeurIPS. This dataset is fairly unique and of extremely high quality, filling a gap that is needed in the Earth science community.

I do not know how other reviewers will feel about this, but I personally find the "dataset first (main text), experiments later (supplementary material)" approach taken by the authors to be refreshing. Too many papers focus solely on benchmark results and fail to adequately describe the dataset and its possible uses. With that said, Table 1 (especially the top half) could probably be moved to the Appendix, with the additional space allocated to Tables 3 and 4 from the Appendix instead.

I would also like to thank the authors for their use of GeoTIFF and shapefile formats as opposed to other more esoteric file formats.

**Additional Feedback:**

Minor comments:

* Some numbers have a space after the thousands space: e.g., 123, 456. Could be a result of using math mode instead of text mode.
* Figure 1: Units like m or km should be separated by a single space (1 m instead of 1m) and non-italic
* Figure 4: There is a (accidental?) set of square brackets [] between the figure and the caption
* Tables: It is recommended to avoid vertical lines when using the booktabs package
* Tables: Not sure if NeurIPS has an official stance on this, but I usually see table captions above and figure captions below

Typos:

* Line 159: acousstic -> acoustic
* Line 170: differencces -> differences

Grammar:

* Line 211: require -> requires
* Line 282: segmentation -> the segmentation
* Line 287: the capability -> with the capability

**Clarity:**

Overall, the paper is extremely clear and well written. At several points of the paper, I was confused about something, only to find that my question was answered in a later section. I don't consider this a flaw of the organization of the paper, and I think it would be foolish to introduce all required background information upfront. For example, I didn't know what a physiographic zone was at first, but it was eventually explained, and I didn't really need to know that information until it was explained. So the rest of my comments are things that I still find unclear, even after reading the full paper.

* The phrase "geographic layers" confused me at first because I primarily think of "geography" as horizontal divisions and "layers" as vertical divisions (e.g., layers of the atmosphere, ocean, or crust), but in reality it is neither. Perhaps a word like "features" would be more clear?
* Why are there more annotated segmentation masks than sonar images?
* I didn't know where Delmarva was. Easy to look up, but also easy to explain in the paper.
* Which interpolation method was used to fill in nodata pixels?
* It would be worth documenting the mapping from 144 descriptions to 9 distinct categories for habitat in the Appendix.
The naming scheme for the paragraphs in section 3.2 is inconsistent. "Input Layers" does not list all input layers, and "Mapping Layers" appears twice.
* It is unclear which mapping tasks are classification and which are segmentation. Fault/fold are described as binary "segmentation", but also as whether an image "patch" contains this feature. Did the authors mean image "pixel" or binary "classification"? I think everything is multiclass or binary segmentation, but it would be nice to clarify this.
* For the SeafloorGenAI portion of the dataset, how many image labels were manually written by marine scientists? All 700K?

**Correctness:**

The preprocessing pipeline and data augmentations performed during dataset construction are sound, and demonstrate careful considerations for domain-specific data peculiarities (such as nodata pixels).

The authors carefully consider in-distribution and out-of-distribution experiments with grouped geographic cross validation. I have no concerns about the experiment setup, although I looked at it in less detail since it's in the Appendix, not the main text.

If you ever expand to global coverage, it may be more useful to use sine and cosine embeddings of latitude and longitude instead of the raw numbers to correctly model the cyclic and non-Euclidean nature of these inputs. But if you're only covering the US, this probably doesn't matter much.

When patchifying the dataset, the authors use a patch size of 224 and a step size of 56 for their sliding window. Although this prevents information loss near the borders, it also results in significant data duplication, leading to a dataset that seems large but is actually 16x smaller than it sounds and takes 16x longer to train each epoch. It also makes it more challenging to compute overall accuracy, as it would be more correct to only consider duplicated pixels a single time instead of simply averaging all pixels. Some libraries like [TorchGeo](https://github.com/microsoft/torchgeo) support raw data (before patchifying), allowing the user to choose random sampling during training and sliding window evaluation during testing. This may be more efficient in terms of storage space, training time, and accuracy calculation.

P.S. As a user of TorchGeo, I would love to see a data loader for your dataset added someday to make it easy to use in a PyTorch training pipeline. However, this can happen after the review period and will not affect my final review score as I am obviously biased.

**Documentation:**

Dataset is well documented, particularly the data provenance and preprocessing. License (CC-BY-4.0), intended uses, availability, and maintenance are also documented. Ethical and responsible use could be better documented (see below). Experiments are also well documented, and all data processing code will be released after review.

USGS and NOAA data is said to be CC0-1.0 and CC-BY-4.0. USGS/NOAA data is usually released as "public domain" without clarification as to which of the many public domain licenses it belongs to. Unless there is explicit documentation of CC0-1.0, I would replace that word with "public domain".

**Ethics:**

Line 237: "Detecting these features is crucial for understanding seismic activity and geological history of the marine environment."

What the authors fail to mention is that detection of faults and folds are also incredibly important to the oil and gas industry (especially chevron folds). However, energy companies likely have significantly better private data sources, meaning the potential for a model trained on SeafloorAI to cause negative environmental impacts is low. Furthermore, the detection of faults and folds is also important for carbon sequestration, counteracting some of the potential negative impacts. If the authors add "oil and gas exploration and carbon sequestration" to line 237, that will likely be sufficient for documentation purposes.

**Limitations:**

The paper lacks a detailed discussion of the limitations of models trained on this dataset, and only describes limitations of the labels themselves. Due to the limited geographic coverage of the dataset, as described in "Opportunities for Improvement", it is unclear if a model trained on this dataset would transfer to different ocean basins around the world. It should be made abundantly clear that this dataset only covers a single country (USA), nearshore regions, and limited latitudes, and that models on this dataset should not be applied to other regions of the world without rigorous validation experiments. Table 3 of the Appendix suggests poor out-of-distribution performance, and can be referred to when mentioning this.

**Opportunities For Improvement:**

The biggest limitation I see of this paper is the geographic extent. Although this paper already provides a much larger and more diverse geographic coverage than previous works, it still misses many important and common bathymetric regions. For example, the dataset only contains nearshore regions, completely omitting open oceans. It is limited to the US and leaves out the Gulf of Mexico, meaning the dataset does not contain coral reefs, one of the most important and diverse ecosystems in our oceans. It contains data from the Atlantic and Pacific Oceans, but not the Indian, Antarctic, and Arctic Oceans, and has very limited latitudinal coverage. From a geologic perspective, the dataset contains both active and passive margins, but no subduction zones. It covers very young and very old oceanic crust, but very little in between.

I realize that datasets like this can take years of field work to create, and asking for more data may not be possible at this stage. However, if the data is already available and only requires additional preprocessing, it may be wise to expand this dataset. If not, then this can be left for future work, as the authors have already mentioned plans to expand the dataset in the future. Specifically, I would focus on the Gulf of Mexico to add coverage of coral reefs and the coast off of Washington and Oregon to add a subduction zone with very different tectonics. Long-term, I would love to see such a dataset expanded to other oceans, especially ones with unique habitats and physiographic zones.

Also see other sections for more minor comments on improvements that can be made to the correctness and clarity of the work.

**Relation To Prior Work:**

The authors provide an extremely detailed and complete comparison with historical work. My only suggestion would be to add more numbers. If a dataset has a small sample size, how small is it compared to your dataset? Also, some citations appear as a long list, such as "[119, 80 110, 37, 117, 54, 63, 12, 115, 114, 62, 126, 111]". I'm not sure if NeurIPS has a specific requirement, but sorting citations by order of appearance rather than alphabetical order would result in nice simple ranges.

**Summary And Contributions:**

In this paper, the authors introduce two new multimodal multitask datasets, SeafloorAI (vision) and SeafloorGenAI (vision-language) for the task of seafloor mapping and understanding. This includes an extensive ML-ready dataset derived from publicly available USGS and NOAA data, as well as captions written by marine scientists. Input layers include sonar backscatter, bathymetry, slope, and rugosity, while output layers include sediment, physiographic zone, habitat, fault, and fold. All output layers are crucial for oceanography and seafloor geology, with a number of wide-ranging applications for better understanding and monitoring the health of our oceans. A vision question answering system trained on SeafloorGenAI provides the ability for marine scientists and other domain experts to analyze and discover relevant information among the ever-increasing "sea" of big data from our oceans.

The main contributions of the paper are the SeafloorAI and SeafloorGenAI datasets. These datasets are both "multi-modal" (several input data sources) and "multi-purpose" (several domain-specific tasks). In total, the "dataset contains 696K sonar images, 827K segmentation masks, 696K general language descriptions and 7M question-answer pairs, covering a total area of 17,300 square kilometers." The authors also introduce their data curation pipeline and release all source code, allowing the dataset to be expanded in the future and allowing similar data curation strategies to be used in other related domains. Although the authors mention developing "a framework that standardizes… nomenclature of geological attributes across data surveys", it is unclear to this reviewer whether this is a novel contribution, or whether they simply borrow existing frameworks (Barnhardt classification and CMECS).

---

> ### Author Rebuttal · Authors · 2024-08-16
>
> **Rebuttal (1/2)**
>
> We thank the reviewer for the valuable comments. Below are our responses to the points raised by the reviewer. We hope these will help improve the clarity of the paper.
>
> **1. Limitation of geographical diversity.**
> First, we thank the reviewer for acknowledging that the hydrographic data surveys could take years to create. Second, we further discuss the diversity in terms of seafloor characteristics that our dataset currently possesses and plans for data expansion.
>
> Our existing dataset, which we see as a starting point towards a goal of more extensive global settings, is very representative of geologically diverse environments. We have data gathered from both an active (West Coast) and passive (East Coast) plate boundary with varied tectonic history and lithologic composition.  Furthermore, within the east coast dataset we have a range spanning from the Middle Atlantic bight with a wide continental shelf all the way up into New England which includes direct glacial impacts on the coastline and seabed. The areas represent similar settings around the world [1, 2]. For instance the East Coast of the US as an Amero-trailing edge tectonic setting is very similar to that found off the east coast of South America. The West Coast of the US, which is an active plate boundary with a narrow shelf, is also similar to those found along the west coast of South American and other similar plate boundary settings. There are admittedly more geological settings that we have hopes of gathering but we must work first with what is available and in this paper we seek to develop and demonstrate the effectiveness of the framework that could be acquired readily at this time.
>
> We are actively collaborating with researchers and experts in the marine science community to expand our dataset by securing access to additional data sources. Specifically, we have had engagement and discussions with Seabed 2030 (https://seabed2030.org/), and with the USGS for access to data in the Gulf Coast. We continue to look for partners to expand our datasets and we hope through this paper that it will help to encourage additional partners to contribute. We also plan to open-source our data processing code so that domain experts can easily contribute their data to our dataset.
>
> [1] Inman and Nordstrom, On the Tectonic and Morphologic Classification of Coasts, Journal of Geology, 1971.
>
> [2] Davis and FitzGerald, Beaches and Coasts, 2019.
>
> **2. Is the dataset useful to the broader machine learning community?**
> While it is true that sonar imagery is less common in the literature compared to more established fields like MRI imaging or remote sensing, this scarcity highlights the importance of expanding machine learning's reach into underrepresented domains. Our dataset specifically addresses this gap, offering a valuable resource for researchers and practitioners who are eager to explore new applications of machine learning. One of the core objectives of ML is to advance scientific applications across diverse areas; therefore, sonar imaging is as valuable as other scientific applications in the realm of ML. By introducing this dataset, we aim to bring greater attention and innovation to the domain of sonar imagery.
>
> **3. How many image labels were manually written by marine scientists?**
> In the language annotation process, the marine scientists manually examined 50 data samples and wrote a detailed description for each one. These samples will serve as the In-Context Learning (ICL) examples [3] for GPT-4 to mimic the domain language upon generating the detailed descriptions and question-answer pairs for the remainder of the dataset. In brevity, ICL is a few-shot learning paradigm for large language models (LLMs), enabling the model to adapt to new tasks with a few examples without the need for explicit retraining. The original ICL paper demonstrated that just one or few examples (ten) could significantly enhance the accuracy of LLMs in new tasks. Therefore, 50 examples manually annotated for ICL in our pipeline is sufficient. Besides, the process of manual annotation was notably time-consuming. This is because the marine scientists must meticulously analyze each sample to accurately describe the image patch across different geological layers.
>
> [3] Brown et al., Language models are few-shot learners, NeurIPS 2020.
>
> **4. Inclusion of the mapping from 144 habitat descriptions to 9 distinct categories.**
> In habitat mapping, there are two main approaches: "lumper" (aggregating categories) and "splitter" (dividing into finer classes). We initially adopted a "lumper" approach, reducing 144 finely discretized habitat types to 9 major classes to address machine learning challenges posed by underrepresented classes. This aggregation improves model reliability without losing information, as the original 144 classes are preserved for detailed analysis. The final paper will include a detailed mapping of these classes, balancing robust model performance with the preservation of detailed habitat data.
>
> **5. Why are there more annotated segmentation masks than sonar images?**
> Five geological layers in our dataset represent five [mapping] tasks. Ideally, if all the sonar images have their corresponding segmentation masks for each geological layer, the number of segmentation masks is 5 times the number of sonar images. However, some of the data we collected does not contain annotations in certain layers, resulting in some missing numbers in the bottom half of Table 1, as mentioned in the Limitation and Future Work section.
>
> **6. Usage of the “geographic” terminology.** We apologize for the confusion; we will change “geographic features” and “geographic layers” to “geological features” and “geological layers”. It is accurate to use “geological” as sediments, physiographic zones, etc. to refer directly to geology and not geography (locations).
>
> For other minor issues, please refer to the follow-up rebuttal.

---

> > ### Author Rebuttal · Authors · 2024-08-16
> >
> > **Rebuttal (2/2)**
> >
> > **7. Clarification for Delmarva.** Delmarva refers to the regions near Delaware, Maryland and Virginia. We will explain where Delmarva is in the final version of the paper.
> >
> > **8. Which interpolation method was used to fill in nodata pixels?** We used nearest neighbor as our interpolation but also plan to try other interpolation methods to see the effect on the experiment results.
> >
> > **9. Clarification on the tasks.** In our paper, we introduce two tasks for the pure vision dataset, SeafloorAI: multiclass and binary segmentation. There is no classification task in the dataset at the moment.
> >
> > **10. TorchGeo integration and additional editorial feedback.** The recommendation to integrate our framework with TorchGeo is valuable, and we will explore this possibility to enhance the accessibility and usability of our dataset in PyTorch training pipelines. We also appreciate the thorough review that identified minor typos, grammatical issues, and formatting suggestions. We will carefully address each of these points in our revision to ensure the highest quality presentation of our work.

---

> > > ### Author Response · Authors · 2024-08-23
> > > **We would like to hear back from Reviewer X3ie**
> > >
> > > Dear Reviewer X3ie,
> > >
> > > We kindly request your feedback on our rebuttal. We've addressed key points including: geological diversity and expansion plans, relevance to the broader ML community, details on our language annotation process, explanation of our habitat mapping approach, and terminology clarifications. We appreciate your time and look forward to your response on these clarifications.
> > >
> > > Best,\
> > > Authors

---

> > > > ### Comment · Reviewer_X3ie · 2024-08-27
> > > > **Increased rating**
> > > >
> > > > Thanks for your detailed follow-up. It seems geographic diversity along the US coastlines is better than I suspected. I also appreciate the authors honesty about missing some types of coastlines and plans to improve the dataset in the future. The biggest concern of other reviewers is the accuracy of the mostly-ML-generated labels. However, I see the images and masks themselves as more impactful, and don't think that SeafloorGenAI detracts from SeafloorAI. As long as the minor comments I left are addressed in the final paper, I see no reason why this paper should not be accepted. I improved my rating (7 -> 8) and wish you luck with the other reviewers.

---

> > > > > ### Author Response · Authors · 2024-08-27
> > > > >
> > > > > Dear Reviewer X3ie,
> > > > >
> > > > > Thank you for your thoughtful follow-up and improved rating. Your feedback has been invaluable in strengthening our work, and we sincerely appreciate your support.
> > > > >
> > > > > Best,\
> > > > > Authors

---

### Official Review · Reviewer_KQU4 · 2024-07-25
**Great dataset but no link provided! Need discussion on geographic extent and could improve impact by including the modeling part**

**Rating:** 7
**Confidence:** 2
**Clarity:** The paper is written in a very detail…

**Review:**

The submitted work is highly significant and has the potential of being an important benchmark dataset for seafloor applications.

The benchmark bridges a gap of existing datasets, between existing ML-ready datasets which suffer from small geographic scales and extensive hydrographic surveys which suffer from inconsistencies across different surveys, e.g. in nomenclature.

The extension of the dataset into the language domain by creating a visual question answering (VQA) dataset, may open a new and innovative way for marine scientists to interact with datasets for analysis and discovery.

The paper is clearly written and the data preparation and processing steps are described in great detail.

Unfortunately, the paper doesn’t succeed to discuss the benchmark modeling as the focus is put on the data processing part and doesn’t leave place for anything else. Yet the authors did run experiments with baseline models which are introduced in the Supplementary material.
I would suggest the authors to consider restructuring the paper by transferring some details of the data preparation into the Supp. Material (for example details of the Mapping Layers in section 3.2).
This way, the authors would be able to enrich the end of the paper which seems to be a bit cut-off (starting with the section on Instruction-following Mapping), adding a section on the modeling with (preliminary) results and including a “proper” conclusion.

Finally, there are some inconsistencies and missing information in the Supplementary Material regarding data and code accessibility and usage. The authors say to provide an URL to a subset of the dataset during the review period – this is missing as far as I can tell. Besides this, I couldn’t find the code for the models on the anonymous github repo.

**Strengths:**

The work demonstrates a big effort to collect and harmonize an extensive dataset across 62 different surveys and 9 regions. One thing which stands-out is the collaboration with domain experts to curate the dataset, e.g. to annotate the geological features and to describe and validate image descriptions and question-answer pairs.

The created multi-modal datasets (consisting of various channels: backscatter, bathymetry, slope, rugosity, coordinates; annotated ground-truth labels of geological features for training segmentation tasks; image descriptions and QA pairs for VLM) is focused on facilitating the development large vision and vision-language foundation models. The data-processing codes which made the curation of this dataset possible is opensourced and available to the community to contribute to the dataset.

**Additional Feedback:**

Line 141: Supplementary -> Supplementary Materials.

Line 146: Geographic feature -> Geological features?!

**Correctness:**

The presented datasets are curated in a sound way. The collection and preparation process are described in high detail in the paper and the processing codes are publicly released.

**Documentation:**

The datasheet for datasets questionnaire is comprehensively answered in the Supplementary Material.

The authors provide an anonymous GitHub repo, featuring their data processing codes under an open CC-BY 4.0 license. However, code of the models (used in the Supp. Material) is not provided.

The authors promise to publicly release their dataset upon acceptance on HuggingFace or Zenodo. They say a link to a subset is provided to the reviewers – unfortunately I couldn’t find any dataset.

**Ethics:**

No ethical concerns are observed.

**Limitations:**

As mentioned above, the dataset is described as geographically diverse, however it only covers regions in the US. This should be discussed:
-	Is this a real limitation of the dataset or are the samples representative for other geographic regions?
-	Can this introduce bias when applying models trained on this dataset on other regions?

The authors share limitations of the dataset they experienced when running experiments with models which is discussed based on the modeling results in the Supplementary materials. The authors should consider moving this discussion into the main paper which will be highly insightful for researchers that want to use the dataset in modeling applications.

**Opportunities For Improvement:**

To increase the impact of the paper, I would enrich the end of the paper which doesn’t make the step from dataset curation into modeling. As I mentioned above, I would suggest the authors to consider reducing some parts of the dataset processing details (e.g. moving parts of 3.2 into the Supp. Material), in order to have space to discuss modeling approaches. Even though only preliminary results are shown, it illustrates an important point how to use the dataset and also offers the opportunity to discuss challenges (as seen from the modeling part in the Supp. Material).

Further points of improvement:
The abstract is somewhat confusing as it talks about two datasets (SeafloorAI and SeafloorGenAI), but then describes the scale of a single dataset. I would suggest introducing SeafloorAI, describe its scale, and then introduce SeafloorGenAI as an extension of the former which enables VL modeling.
Also, the title of the paper only refers to a single dataset.

After reading abstract and introduction, I didn’t expect that the described “geographically diverse” dataset only covers US regions. It would be appreciated if the authors could briefly discuss why the dataset can be considered as being diverse (in terms of seafloor characteristics) and if the regions are representative for other regions globally to which would allow generalization/transfer learning when models are being applied in other geographic regions. This can be added explicitly to line 137/138.

Geographic layers/features (Line 6, Caption Figure 1, line 40, line 46, line 264, etc.) are interchangeably used with geological features (Line 135). I was confused about the geographic layers until the geological features were introduced in section 3.1. Does it make sense to change all “geographic layers” into “geological layers”?

What is unclear is how the different resolutions from 1m to 10m of the rasterized data are integrated? Does each region has its unique resolution? Does this mean it’s not straightforward to mix data from different regions?

**Relation To Prior Work:**

The submissions discusses related datasets, their draw-backs and limitations and argues, based on this, how the presented dataset fills a gap in the literature.

**Summary And Contributions:**

The authors present a large, ML-ready benchmark dataset for seafloor mapping application. The work entails an extensive collection of sonar imagery and derived variables thereof, originating from various surveys and different geographic regions. A standardization framework is developed which harmonizes the different data sources.
Furthermore, the authors establish a second, related dataset adding language components to allow for Vision-Language Modeling.

---

> ### Author Rebuttal · Authors · 2024-08-16
>
> We thank the reviewer for the valuable comments. Below are our responses to the points raised by the reviewer. We hope these will help improve the clarity of the paper and help the reviewer to finalize the judgment.
>
> **1. Limitation of geographical diversity.**
> First of all, it is important to recognize that public hydrographic surveys that meet our specific needs are scarce, largely due to the considerable time and effort required for recording sonar images and annotating segmentation masks. Reviewer ``X3ie`` has acknowledged that datasets like ours can take years of field work to create.
>
> Our existing dataset, which we see as a starting point towards a goal of more extensive global settings, is still very representative of geologically diverse environments. We have data gathered from both an active (West Coast) and passive (East Coast) plate boundary with varied tectonic history and lithologic composition.  Furthermore, within the east coast dataset we have a range spanning from the Middle Atlantic bight with a wide continental shelf all the way up into New England which includes direct glacial impacts on the coastline and seabed. The areas represent similar settings around the world [1, 2]. For instance the East Coast of the US as an Amero-trailing edge tectonic setting is very similar to that found off the east coast of South America. The West Coast of the US, which is an active plate boundary with a narrow shelf, is also similar to those found along the west coast of South American and other similar plate boundary settings. There are admittedly more geological settings that we have hopes of gathering but we must work first with what is available and in this paper we seek to develop and demonstrate the effectiveness of the framework that could be acquired readily at this time. Our goal is to develop and test this framework and by illustrating the effectiveness and benefits in using this framework to encourage further adoption to expand the representative datasets.
>
> We are actively collaborating with researchers and experts in the marine science community to expand our dataset by securing access to additional data sources. Specifically, we have had engagement and discussions with Seabed 2030 (https://seabed2030.org/), and with the USGS for access to data in the Gulf Coast. We continue to look for partners to expand our datasets and we hope through this paper that it will help to encourage additional partners to contribute. We also plan to open-source our data processing code so that domain experts can easily contribute their data to our dataset.
>
> [1] Inman and Nordstrom, On the Tectonic and Morphologic Classification of Coasts, Journal of Geology, 1971.
>
> [2] Davis and FitzGerald, Beaches and Coasts, 2019.
>
> **2. How are different resolutions integrated?**
> Our dataset contains multiple resolutions, with each region having one resolution. We provide a table here about such details and will include it in the bottom half of Table 1 in the final version:
>
> | Region        | Resolution |
> |---------------|------------|
> | 1, 2, 3, 4, 8 | 2m/pixel   |
> | 5             | 10m/pixel  |
> | 6,7           | 1m/pixel   |
> | 9             | 4m/pixel   |
>
> We keep the original resolutions of the data as collected from the public data survey so that the users can select the resolution that fits their applications. For example, a user might utilize coarser resolutions in seafloor mapping as a larger scale, and finer resolutions in fine-grained object detection, such as crab pots, shipwrecks, etc. Although our dataset currently does not include any object detection task, it can be used to pretrained self-supervised models to further boost the accuracy of downstream tasks in smaller datasets.
> We recognize that mixing data from different regions with varying resolutions could present challenges. One way to synchronize the data resolution across regions is via resampling to the coarsest resolution. Another way is to apply multi-resolution deep learning techniques to handle different resolutions from various regions. Ultimately, our work aims to develop a foundation model(s) that could handle various resolutions that can be applied to multiple applications. Therefore, having multiple resolutions in our dataset offers great flexibility.
>
> **3. Inclusion of modeling.**
> Due to space limitations in the submission phase, we decided to put more emphasis on the curation of the data and its usage, hence focusing less on modeling. Upon acceptance, with an extra page limit, we will move the top half of Table 1 to the supplementary material to make room for the experiments.
>
> **4. Terminology usage: geographic vs. geological.** We apologize for the confusion; we will change “geographic features” and “geographic layers” to “geological features” and “geological layers”. It is accurate to use “geological” as sediments, physiographic zones, etc. to refer directly to geology and not geography (locations).
>
> **5. Working Zenodo link.** We apologize for the oversight on the Zenodo link. We have fixed the issue and provide the working link here: https://zenodo.org/records/11630750. These are the samples from Region 5 in the paper. In the anonymous git repository (https://anonymous.4open.science/r/SeafloorFM), we also added a script to visualize the data (``visualize_vision_data.py`` and ``visualize_vlm_data.py``) and some visualization of the data (under ``vis-vision/`` and ``vis-vlm/`` directories) for the reviewers’ convenience (as the data for this region takes up 40GB of storage). The model training code is also included.
>
> **6. Clarifying dataset distinction and abstract structure.**
> We appreciate your suggestion and will revise the abstract to clearly distinguish SeafloorAI and SeafloorGenAI, presenting SeafloorAI first with its scale, followed by SeafloorGenAI as its vision-language extension, and we will update the title to reflect both datasets.

---

> > ### Author Response · Authors · 2024-08-23
> > **We would like to hear back from Reviewer KQU4**
> >
> > Dear Reviewer KQU4,
> >
> > We kindly request your feedback on our rebuttal. We've addressed key points including: geological diversity, clarification on data resolutions, plans to include modeling details in the main paper, terminology corrections, and provided a working dataset access link. We look forward to your thoughts on these clarifications and appreciate your time.
> >
> > Best,\
> > Authors

---

> > > ### Author Response · Authors · 2024-08-28
> > >
> > > Dear Reviewer KQU4,
> > >
> > > We want to follow up on our previous reminder regarding our rebuttal.
> > > All other reviewers have responded and Reviewers ``47bb``&``X3ie`` have increased their ratings.
> > > We would greatly appreciate your thoughts on our clarifications. Thank you for your time and consideration.
> > >
> > > Best,\
> > > Authors

---

> > ### Comment · Reviewer_KQU4 · 2024-08-30
> > **Rebuttal response**
> >
> > Dear authors,
> > I sincerely want to apologies for my delayed response to your rebuttal which you provided in a very timely manner.
> > I would like to thank the authors for taking my feedback constructively and addressing all my concerns. In particular I appreciate the update of the zenodo link, the inclusion of the modeling codes in the github repo, and the promised restructuring of the paper for the camera-ready version to include modeling approaches and baseline results. I acknowledge the scarcity of hydrographic surveys and recognize the authors efforts for collaboration and their contribution to the domain by publishing the data processing code for researchers to more easily add datasets.
> > With this, I will increase my rating and support the acceptance of the paper for publication.

---

> > > ### Author Response · Authors · 2024-08-30
> > > **Thanks for Increasing the Rating!**
> > >
> > > Dear Reviewer KQU4,
> > >
> > > Thank you for your thoughtful response and for taking the time to review our rebuttal. We greatly appreciate your understanding regarding the challenges in hydrographic survey data collection and your recognition of our efforts to address this through collaboration and open-sourcing our data processing code. Your feedback throughout this process has been invaluable in helping us improve our work.
> > >
> > > Best,\
> > > Authors

---

### Official Review · Reviewer_47bb · 2024-07-29
**SeafloorGenAI.  Lots of images and a few human annotations**

**Rating:** 7
**Confidence:** 3
**Correctness:** yes
**Clarity:** yes

**Review:**

I have mixed feelings about this paper, and for some parts of the paper analysis, would defer to others that have more experience with SONAR.  The dataset generated is orders of magnitude larger than existing datasets (to my knowledge) and appropriate to support some of the more modern approaches to vision/language modeling.  The paper discusses important and useful ways that the dataset is organized/interpolated etc. to make it consistent.

I am somewhat less compelled by the language annotation.  “only” 50 scenes are directly annotated by domain experts.  The remaining annotations are generated by GPT-4 prompts — but the examples shown in the paper and in the json file of 100 example annotations on the datasharing site make these descriptions basically listing facts from the extracted data.  If a vision algorithm says that an image is 48.75% Continental/Island Shore Complex with the following polygon structure, it isn’t at all clear to me that putting that into “sentence form” is useful.  Most of the sentences seem to have this form.

**Strengths:**

While sonar is not something I’ve worked with, I’ve done substantial work in creating large scale datasets for ML from imperfect initial data.  This paper make a large collection of well justified and reasonable choices to make the dataset easier for the ML community to use.

**Additional Feedback:**

none

**Documentation:**

yes

**Limitations:**

yes

**Opportunities For Improvement:**

1. I would like to understand the potential real use case for the language annotations, instead of having the extracted data in a table and running queries against it.
2. I would like to better understand how the expert based “text annotation checking” happened (apologies if I missed it).  What fraction of annotations had to be corrected?  of the 696K images, were all of the annotations checked?   Were there any interesting outliers found?

**Relation To Prior Work:**

yes

**Summary And Contributions:**

This paper addresses the gap in the availabiliyt of large scale sonar datasets to support ML training.
They compile public hydrographic surveys, and worked to unify nomenclature across these datasets to compile 	696K side-scan sonar images and 827K segmentation masks for 5 geographic layers: Sediment, Physiographic Zone, Habitat, Fault, and Fold.
Marine science expertise was used to describe 50 randomly selected samples from the dataset.

GPT-4 is then used to create language descriptions, and to generate question-answer pairs about the rest of the dataset (the other 695.95K) images based on features extracted from the data (distribution of backscatter, slope, depth, composition percentages, and their relative layouts).

---

> ### Author Rebuttal · Authors · 2024-08-16
>
> We thank the reviewer for the valuable comments. Below are our responses to the points raised by the reviewer. We hope these will help improve the clarity of the paper and help the reviewer to finalize the judgment.
>
> **1. Why are only 50 scenes (or image patches) annotated by experts?**
> As outlined in the paper, we utilized GPT-4 to generate language annotations for all samples, leveraging In-Context Learning (ICL) [1] during the language generation process. In brevity, ICL is a few-shot learning paradigm for large language models (LLMs), enabling the model to adapt to new tasks with a few examples without the need for explicit retraining. The original ICL paper demonstrated that just one or few examples (ten) could significantly enhance the accuracy of LLMs in new tasks. Therefore, 50 examples manually annotated for ICL in our pipeline is sufficient. Besides, the process of manual annotation was notably time-consuming. This is because the marine scientists must meticulously analyze each sample to accurately describe the image patch across different geological layers. It is important to note that these 50 expert annotations are just the initial set, and we are committed to expanding this number to further enhance the quality and reliability of our dataset.
>
> [1] Brown et al., Language models are few-shot learners, NeurIPS 2020.
>
> **2. What is the purpose of language annotations?**
> Our paper focuses on curating a dataset designed to support the development of large vision-language models that can assist domain experts in the analysis process. To achieve this, we create analysis-driven question-answer pairs that center around important geophysical parameters and segmentation masks of specific geological features within an image patch. While the geophysical parameters can be calculated from the input layers, the segmentation masks are manually annotated by marine scientists. With a model trained on our dataset, experts can efficiently query features of interest within the image patch and derive meaningful insights from the resulting information.
>
> As highlighted in the Limitation and Future Work section, our long-term goal is to develop an enhanced version of the current dataset that incorporates reasoning capabilities tailored to marine science applications, one of which is survey exploration. For instance, when exploring survey data, marine scientists might need to identify regions with specific geological features across a vast survey area. They could prompt the model with queries like, “Identify areas suitable for offshore wind development” or “Find regions with coarse sand and ripples within a divable depth range suitable for the placement of artificial reefs.” To enable the model to respond to such inquiries, the language annotations must involve reasoning capabilities, allowing the trained model to accurately identify areas with geological features that meet criteria such as “offshore wind development” or “placement of artificial reefs.”
>
> **3. How are the language annotations validated?**
> In the language annotation pipeline, we asked the marine scientists to validate a thousand samples generated by GPT-4, especially the generated language descriptions. The experts are tasked to validate the descriptions based on two criteria: (1) consistency to the original annotations and (2) coherency to the domain language. Currently, we are on the last phases of quality assessment and will include the details on the evaluation pipeline as a separate section in the Supplementary Materials. We further describe our language annotation procedure as follows:
>
> Due to budget constraints on the usage of OpenAI API, we meticulously designed our procedure with multiple iterations. The idea is to engineer our prompt to GPT-4 on a small subset of data before applying it to the whole dataset. For each iteration, (1) we annotated a thousand random samples with GPT-4, (2) the marine scientists reviewed the quality of the generated annotations and gave feedback based on the two mentioned criteria, (3) we refined our prompts to GPT-4, a.k.a prompt engineering, to achieve higher quality language annotations, (4) we repeated the steps for the next iteration. Finally, when the quality is met, we will populate the entire dataset with language annotations.
>
> To prevent hallucination in large language models, we made sure to provide GPT-4 as much textual information extracted from the annotations as possible. An example can be found here: https://anonymous.4open.science/r/SeafloorFM/sample_prompt.txt

---

> > ### Comment · Reviewer_47bb · 2024-08-19
> > **I appreciate your rebuttal, and want to follow up.**
> >
> > I feel like some of the responses don't directly address my question.
> >
> > I asked for "what are the real use cases for the text annotations language annotations, instead of having the extracted data in a table and running queries against it."
> >
> > ... and the response was to train a large language model on it.  I don't like that answer.  Large language models are a tool not an end use.  What is the end use that the LLM tool will support?
> >
> > I don't think it is general question and answer about these images --- LLMs do really well with they are trained with massive data written by people with different perspectives and different expertise.  You have created millions of sentences about image that say:
> > "this is an image with 48% continental shore complex..."
> > "this is an image with 51% continental shore complex..."
> >
> > I certainly understand the need to develop these tools step by step, but it is less clear to me that the current step has so much value.
> > "this is an image with 41% continental shore complex..."
> >
> > I wonder if you did (and apologies if I misssed it in the paper) any follow up evaluation ... after all the steps of iterative refinement with the experts, any evaluation of the final model.  Something like here are the final descriptions that we make for 50 unseen (by the model) images, how often would an expert agree with the analysis?  How often would they say that some important feature was missed?
> >
> > There might be a partial answer to this already in the iterative training step (something like: "by the fourth iteration, expert reviewers only felt the need to update 3% of the annotation)?

---

> > ### Author Response · Authors · 2024-08-21
> >
> > Dear Reviewer 47bb,
> >
> > We apologize for any confusion in our initial response and appreciate the opportunity to clarify the utility of language annotations and how we generated them in our study of oceanographic datasets.
> >
> > To begin with, we did not train any LLM. The 50 samples that marine scientists manually annotate serve as the examples for In-Context Learning (ICL). To give more details on ICL, it is a prompt engineering technique that enables a model to adapt to new tasks by providing examples within the prompt, allowing the model to generate relevant responses based on these examples without the need for additional training. In this case, the GPT-4’s “new task” is to generate the general descriptions and question-answer pairs for our dataset.
> >
> > Regarding the utility of our language annotations, the oceanographic datasets are inherently complex and spatially dense, requiring significant post-processing and expertise to convert raw data into actionable information. By employing a large model (LLMs or large VLMs) trained on our dataset, we can directly query processed data products, bypassing the labor-intensive step of creating and maintaining queryable tables for each dataset. This significantly streamlines the workflow and reduces the expertise required for data interpretation.
> >
> > Moreover, these large models can provide practical benefits to end-users. Take a fisheries biologist as an example. The biologist may have a species of interest which they would like to target. The species in question will have certain habitat requirements that will make certain areas more conducive for sampling efforts compared to others - i.e., depth ranges of 3-10 m, prefer loose gravel substrates adjacent to rocky outcrops, and be close enough to the field operations center. Rather than pulling each dataset into a GIS and filtering the datasets by habitat and spatial requirements, the biologist could simply ask the model to return locations which meet the requirements, negating the need to perform any analysis, and expediting the field sampling.
> >
> > We also recognize that our current annotations primarily provide factual descriptions—such as the percentage coverage of various seabed types. These serve as a foundation for more sophisticated, reasoning-based annotations for future improvements. For the “Identify areas suitable for offshore wind development” example (mentioned in the rebuttal), the model would need to find flat sandy portions of the seabed within 15-40 m of depth in close distance to the shoreline, which ties back to the factual descriptions that we currently have in this version of the annotations.
> >
> > Next, we would like to provide some initial results on the human evaluation of the language annotations. As mentioned before, we conduct this procedure in multiple iterations on a random subset of samples with the goal to refine our prompt to GPT-4 so that it incrementally generates higher quality annotations.
> >
> > We define the metrics based on facts and coherence as follows:
> > 1. **Factual consistency:** the information of geological features in the descriptions are consistent with the raw data.
> > 2. **Factual completeness:** the coverage of geological features in the descriptions.
> > 3. **Coherence:** the logical and fluid integration of information in the text, allowing it to be easily understood by the readers. This metric is qualitative.
> >
> > For example, during Iteration 2, the **factual consistency** is perfect, but some descriptions miss information about certain geological features, as depicted by **factual completeness** in the table below:
> >
> > | Features| Factual Consistency | Factual Completeness
> > |-------------|---------------------|------
> > | Backscatter| 100%| 96%
> > | Bathymetry| 100%| 96%
> > | Slope| 100%| 96%
> > | Rugosity| 100%| 96%
> > | Coordinates| 100%| 98%
> >
> > Regarding **coherence**, listed below is the general feedback from the marine scientists:
> > - While the description contains the information of interest, the output is not as eloquent as the human annotation. Rather than reading as a description of an area as you would see in a textbook or other educational resource, the output reads more as a list: “Bounds of image, Top-left…..; The shallowest depth is…..; The rugosity a mean intensity of….;”. Rather than “This sonar image represents an oceanographic area bounded by coordinates….; The depth ranges from….; Rugosity indicates a very uniform surface….”.
> > - The output lacks keywords which help to distinguish the context of the image. Missing words include “sonar image”, “oceanographic”, “coordinates”, “geophysical and geological data”.
> >
> > Based on such feedback, we revised our prompt by adding necessary guidance for GPT-4 to make sure all information from the raw annotations are covered and create more coherent descriptions that align with domain standards. For instance, we created a list of essential domain-specific keywords and incorporated these keywords into the prompts to encourage their use in the generated descriptions.

---

> > > ### Comment · Reviewer_47bb · 2024-08-27
> > > **thank you for the response**
> > >
> > > I appreciate the thoughtful responses.  I remain uncertain of the long term usefulness of the text annotations; but also believe that papers do not have to answer all possible questions to be a good contribution.  I think the authors have put together a very good, interesting dataset for a problem domain where those datasets have not previously been as available or well organized.  I've increased my rating to Accept.

---

> > > > ### Author Response · Authors · 2024-08-27
> > > >
> > > > Dear Reviewer 47bb,
> > > >
> > > > Thank you for your thoughtful consideration of our rebuttal. We appreciate your acknowledgment of the dataset's value and organization, especially in a domain where such resources have been scarce. Your increased rating and support for acceptance mean a great deal to us. Thank you for your valuable feedback throughout this process.
> > > >
> > > > Best,\
> > > > Authors

---

### Author Response · Authors · 2024-08-16
**General Response**

To all reviewers, we would like to thank you for your very detailed and valuable feedback. We appreciate the reviewers find our work is of ``"high quality and originality"`` (X3ie, AnUV) with ``"big effort"`` (KQU4) in collecting very large-scale data, ``"highly significant"`` (KQU4), ``"bridges a gap of existing datasets"`` (KQU4), has ``"potential impact of ocean stewardship"`` (X3ie), and ``"fills a gap that is needed in the Earth science community"`` (X3ie); our dataset is ``"well-justified, reasonable"`` (47bb), ``"significant to the marine science"`` (AnUV), is ``"the largest dataset for sonar images that is made publicly for machine learning use"`` (AnUV), ``"cover multiple use cases"`` (AnUV), and has ``"the potential of being an important benchmark dataset for seafloor applications"`` (KQU4); our paper is ``"extremely well written and flows well"`` (X3ie, KQU4, 47bb).

**Contribution of this work.** We propose a large-scale, multi-task, multimodal (vision and language) seafloor mapping dataset. The dataset consists of 62 geo-distributed data surveys across 17,300 square kilometers, with 696K sonar images, 827K annotated segmentation masks, and approximately 7M question-answer pairs. In addition, we provide a standardization of naming convention across these surveys, under the rigorous supervision of marine scientists, to unify an extensive AI-ready dataset. The curation of our dataset is under the supervision of marine scientists to ensure high quality and accuracy.

**Summary of reviews.** Out of four reviewers, Reviewers 47bb, X3ie, and AnUV vote for acceptance. We summarize the concerns of the reviewers below:

1. **Restructuring the paper and data accessibility.** (KQU4, AnUV) We will restructure the paper to include more details on modeling approaches and preliminary results in the main text. We apologize that the Zenodo link was not working properly; here is the working link: https://zenodo.org/records/11630750. These are the samples from Region 5 in the paper. In the anonymous git repository (https://anonymous.4open.science/r/SeafloorFM), we also added a script to visualize the data (``visualize_vision_data.py`` and ``visualize_vlm_data.py``) and some visualization of the data (under ``vis-vision/`` and ``vis-vlm/``directories) for the reviewers’ convenience (as the data for this region takes up 40GB of storage). The model training code is also included.

2. **Use case for language and language annotation validation.** (47bb, AnUV) Language annotations enable natural language querying of complex geological features, facilitate cross-modal learning, and allow experts to interact with the model more intuitively.
We employed a rigorous iterative process for generating and validating language annotations. Marine scientists are asked to validate 1,000 GPT-4 generated samples based on consistency with original annotations and coherency with domain language. Currently, we are on the last phases of quality assessment and will include the details on the evaluation pipeline as a separate section in the Supplementary Materials.

3. **Geographic diversity.** (KQU4, X3ie). Our dataset, though currently limited to US regions, represents diverse geological environments. It includes both active and passive plate boundaries, varied tectonic histories, and lithologic compositions analogous to global settings. The US East Coast mirrors South America's east coast, while the West Coast resembles other active plate boundaries worldwide. We aim to expand to more global settings in the future, using this initial dataset as a foundation to encourage broader adoption and dataset expansion.

---

### Decision · Program_Chairs · 2024-09-26

**Decision:**

Accept (Poster)

**Comment:**

This submission introduces a large-sale sonar datasets for machine learning training. The dataset consists of 62 geo-distributed data surveys across 17.3k square kilometers of area including 696k sonar images and 827k annotated segmentation masks for 5 geographic layers. The provided dataset was enriched by language descriptions and question-answer pairs generated by ChatGPT.

This submission received four thoughtful and in-depth reviews with an **average rating of 7.25**. The authors have responded to the reviews with a detailed rebuttal and the subsequent discussion between the reviewers and the authors was able to address questions and concerns.

Based on the reviewers' feedback, the rebuttal, the discussion, and overall rating, I recommend the paper for **acceptance**. I would like to ask the authors to work through all recommended improvements as given by the reviewers for the final version of the paper.